# Phytochemical and biological investigations on *Centranthus kellereri* (Stoj., Stef. & T. Georgiev) Stoj. & Stef. and *C. ruber* (L.) DC. and their potential as new medicinal and ornamental plants

**Valtcho D. Zheljazkov**[1☯]*, **Ivanka B. Semerdjieva**[2,3☯], **Daniela Borisova**[4☯],
**Elina Yankova-Tsvetkova**[3☯], **Lyubka H. Koleva-Valkova**[5☯], **Galya Petrova**[3☯],
**Ivayla Dincheva**[6☯], **Fred Stevens**[7☯], **Wenbin Wu**[7☯], **Tess Astatkie**[8☯], **Tanya Ivanova**[9☯],
**Albena Stoyanova**[10☯], **Anatoli Dzhurmanski**[11☯]

1 Department of Crop and Soil Science, Oregon State University, Corvallis, OR, United States of America,
2 Faculty of Agronomy, Department of Botany and Agrometeorology, Agricultural University, Plovdiv,
Bulgaria, 3 Department of Plant and Fungal Diversity, Division of Flora and Vegetation, Institute of
Biodiversity and Ecosystem Research, Sofia, Bulgaria, 4 Administration of Vrachanski Balkan Nature Park,
Executive Forest Agency, Ministry of Agriculture, Food and Forestry, Vratsa, Bulgaria, 5 Department of Plant
Physiology, Biochemistry and Genetics, Agricultural University, Plovdiv, Bulgaria, 6 Department of
Agrobiotechnologies, Agrobioinstitute, Agricultural Academy, Sofia, Bulgaria, 7 Department of
Pharmaceutical Sciences and the Linus Pauling Institute, Linus Pauling Science Center, Oregon State
University, SW Campus Way, Corvallis, OR, United States of America, 8 Faculty of Agriculture, Dalhousie
University, Truro, NS, Canada, 9 Department of Technology of Fats, Essential Oils, Perfumery and
Cosmetics, University of Food Technologies, Plovdiv, Bulgaria, 10 Department of Tobacco, Sugar, Vegetable
and Essential Oils, Perfumery and Cosmetics, University of Food Technologies, Plovdiv, Bulgaria,
11 Institute of Roses, Aromatic and Medicinal Plants, Kazanlak, Bulgaria

☯ These authors contributed equally to this work.
* Valtcho.Jeliazkov@oregonstate.edu

e0293877. https://doi.org/10.1371/journal.
pone.0293877

Para, BRAZIL

## Abstract

### Introduction

*Centranthus kellereri* is a Bulgarian endemic plant species, found only in two locations in the
world: The Balkans Mountains (Stara Planina), above the town of Vratsa, and The Pirin
Mountains, above the town of Bansko, Bulgaria. Being endemic and endangered species
precluded any significant research on it. The hypothesis of this study was that the popula-
tions of *C. kellereri* may represent genetically, phytochemically, and morphologically distinct
forms and these will differentiate from *C. ruber*. Furthermore, *C. kellereri* possibly imperfect
embryology may preclude its more widespread distribution under natural conditions.

### Results

This study revealed the phytochemical profile, antioxidant activity, embryology, surface
microstructural morphological traits, and genetic differences between the *C. kellereri* plants
from the only two natural populations and compares them to the ones of the related and bet-
ter-known plant *C. ruber*. The essential oil (EO) content in aboveground plant parts and in

**Data Availability Statement:** All relevant data are within the paper and its Supporting Information files.

**Funding:** This research was supported by Oregon State University startup funds awarded to Dr. Valtcho D. Jeliazkov (Zheljazkov).

**Competing interests:** The authors have declared that no competing interests exist.

roots was generally low and the EO composition varied significantly as a function of plant part, year of sampling, location, and species. Methylvaleric acid was a major EO constituent in the *C. kellereri* EO, ranging between 60.2% and 71.7% of the total EO. The EO included monoterpenes, sequiterpenes, long-chain alkanes and fatty acids. Phytochemical analyses of plant tissue revealed the occurrence of 32 compounds that were tentatively identified as 6 simple phenolics, 18 flavonoids, 1 quinone, 1 lipid, 1 alkaloid, 2 diterpenes, and 3 triterpenes. There were differences in detected compounds between the *C. kellereri* plants at the two locations and between the roots and shoots in both species. The total phenols and flavonoids varied between the two species but were also dissimilar between the plants from the two populations of *C. kelleri*. Free radical scavenging activity, measured with ABTS and DPPH in aqueous and methanol extracts, had similar values; however, overall, *C. kellereri* from Vratsa showed the highest antioxidant activity while *C. ruber* had the lowest activity. Genetic analyses showed a clear differentiation between *C. kellereri* and *C. ruber*, and between the two populations of *C. kellereri*. Embryological studies revealed the peculiarities of the male and female generative spheres of the two species that were defined as being sexually reproducing. The pollen had high viability; however, the low viability of seeds demonstrated possible high sensitivity of *C. kellereri* to the environmental conditions, perhaps the main factor modifying and restricting the population sizes. The SEM analyses exposed differences in surface microstructural traits between the species (C. *kellereri* and *C. ruber*) but also between the two populations of *C. kellereri*. The observed dissimilarities in genetic makeup, micromorphological characteristics, and phytochemical composition strongly indicate that the two populations can be classified as distinct subspecies or varieties of *C. kellereri*; var. *pirinensis* and var. *balkanensis*. Further research is needed to introduce *C. kellereri* into culture and develop it as a high-value specialty crop or ornamental in order to conserve *C. kellereri* natural populations. *C. kellereri* may be utilized as a source for phytochemicals of interest and as an ornamental plant like *C. ruber*; however, it may have a greater environmental plasticity and adaptation as evidenced by its current locations.

## 1. Introduction

*Centranthus kellereri* (subfamily Valerianaceae of Caprifoliaceae Juss.) is an endemic plant species found exclusively at two locations in the world. The first location is in the Balkan Mountains (Stara Planina), situated between the village of Chelopek and the village of Pavolche above the town of Vratsa, Bulgaria [1]. The second location is in the Pirin Mountains, in the areas of Banski Suhodol, the Caves, and Academica locations, located above the town of Bansko, Bulgaria [2].

The two habitats of *C. kellereri* are characterized by plants spreading over mobile sandstone, rocky screes, with a northwestern exposure and an altitude ranging from 700 m in the Balkan Mountains to 1,800 m in the Pirin Mountains, under extreme conditions. The species is a calcicole, with distinct mesoxerophytic features and is considered as a pioneer in the vegetation cover of mesoxerophytic grassland formations. According to the EUNIS classification [3], the habitat of the species is H2.4 Mountainous sandstone and ultrabasic screes in the temperate zone. According to the Habitat Directive, it corresponds to natural habitat 8120 sandstone screes and screes of sandstone shales from montane to alpine levels (*Thlaspietea rotundifolii*) [4].

This species is protected and included in Appendix 3 of the Law on Biological Diversity, as well as in the new edition of the Red Book of Bulgaria, Volume I. Plants and fungi, with the category of Critically Endangered [CR B2ab(v)] [5]. Due to its status as an endemic and endangered species, there has been very little research conducted on *C. kellereri*. Consequently, there are almost no publications on its phytochemistry, including the essential oil (EO), micromorphological traits, genetics, embryology, or bioactivity. Furthermore, the intriguing phenomenon of *C. kellereri*'s restricted distribution to only two locations in two separate mountains, with an air distance of approximately 200 km between them, remains largely unexplained. Understanding the factors influencing this unique distribution pattern could potentially be linked to the species' embryology and reproductive strategies. Unfortunately, data on the features of the structures and running of the processes in the male and female generative spheres in the species of the *Centranthus* genus are limited and partial [6], and *C. kellereri* embryology has not been studied.

The closely related species *C. ruber* has been extensively studied [7–11]. There were several reports on its compositions, bioactivity and potential health benefits [7–11]. However, not much is known about its phytochemistry. *C. ruber* is a well-known and highly valued attractive ornamental plant that has a wide geographic and environmental range of growth, with wide adaptability, and has even triggered concerns regarding its invasive potential [12, 13]. Both species are perennial herbaceous plants with glabrous stems that can reach up to 50–120 cm in height [14]. The stems are simple or branched in the upper half and are upright or ascending. The seeds of both species are one-seeded with a kite of feathery bristles [14].

Key features that distinguish *C. kellereri* from closely related species are the height of the plants, which can reach up to 120 cm [14]. Since the species forms multiple equal stems from its base, the overall appearance of a single plant is like a tuft. The stems and leaves are bare and gray-green [14]. The flowers are purplish-red to pink with tubular corollas ending in a linear spur. The corolla, together with the spur, is 2–2.5 cm long [14]. In Bulgaria, there are no closely related species, but in the global flora, *C. kellereri* is sometimes described as a subspecies of *C. longiflorus* Steven, specifically *C. longiflorus* subsp. *kellereri* (Stoj., Stef. & Georgiev) I. Richardson [15–17]. According to one rare palynomorphological investigation of *C. kellereri*, *C. ruber*, and *C. longiflorus*, it is clear that pollen grains of *C. kellereri* have a larger size, narrower colpi, and thicker exine than those of *C. ruber* and *C. longiflorus* [18]. The results of the present study support the recognition of *C. kellereri* as a separate species rather than a subspecies of *C. longiflorus* [18]. Indeed, according to the official flora of Bulgaria, *C. kellereri* is a Bulgarian endemic species [14].

Natural products have proven to be a significant source of new compounds utilized in new drug development; however, their full potential has yet to be realized [19]. Endemic and protected plant species are often inadequately studied and underutilized as sources for natural products. If a natural product of interest is identified in an endemic species, it can be introduced into culture, and agronomic management practices can be developed for its commercial production. Such an approach may provide several tangible benefits: (1) provides new sources for natural products of interest; (2) can serve as a source of income for human populations living in areas with marginal lands, and (3) aids in the conservation of their often very limited plant populations.

Positive examples of this approach can be seen with Edelweiss (*Leontopodium alpinum* Cass.), a protected and rare species in European mountains, and, is the national flower of Switzerland. The species was domesticated and is being grown as a high-value crop in Switzerland for the production of extracts used in various consumer specialty products such as candy, drinks, and milk [20, 21]. Indeed, a previous study showed that *C. longiflorus*, a closely related species to *C. kellereri*, possesses sedative and anticonvulsant effects similar to those exhibited by diazepam [22].

The hypothesis of this study was that populations of *C. kellereri* may represent genetically, phytochemically, and morphologically distinct forms of varieties. These individuals may be genetically and chemically different from the well-known *C. ruber*. Furthermore, *C. kellereri's* possibly imperfect embryology may preclude its more widespread distribution under natural conditions. *C. kellereri* could serve as a desirable source for phytochemicals of interest and may have ornamental values.

The objectives of this study were to reveal: (1) the phytochemical and genetic differences of *C. kellereri* individuals from the two locations and compare them to the ones of *C. ruber*; (2) explore microstructural differences between the *C. kellereri* plants from the two populations using Scanning Electron Microscopy (SEM); (3) reveal the embryological characteristics of the species from the two populations as a possible explanation for the species limited distribution and endemicity.

## 2. Materials and methods

### 2.1. Plants materials

The plant materials for this study were *C. kellereri* sampled from the only two natural populations: (1) Banski Suhodol located in the northern part of Pirin Mountains (41˚47′56.4′′N, 23˚24′54.1′′E, 2017masl), above the town of Bansko, and (2) Protected area "Vegdata" located in the Vrachanski Balkan Nature Park (43˚08′22.67′′N, 23˚35′59.74′′, 730masl of Stara Planina, (the Balkan Mountains), above the town of Vratsa.

As noted, *C. kellereri* is a critically endangered and protected Bulgarian endemic species under the Biodiversity Act [5]. Therefore, the research team obtained a special permit from the Ministry of Environment and Water (MEW) numbered 736/12.032018 issued to Dr. V.D. Jeliazkov (Zheljazkov), Mrs. Daniela Borisova, and Dr. C. Radoukova. *C. ruber* was grown *ex-situ* and collected from the experimental field of the Institute of Roses, Essential Oil and Medical Crops, in the town of Kazanlak, Bulgaria (42˚38′0.05′′N, 25˚23′0.17′′E, 395masl). Representative subsamples were taken from *C. kellereri* individuals from the two populations and *C. ruber* specimens from the Institute of Roses, Essential Oil and Medical Crops, Kazanlak for DNA, phytochemical, metabolomic, antioxidant, anatomical (SEM), and embryological studies.

Plants were sampled for DNA, phytochemical, and some embryological analyses in May-June and August-September during two subsequent years. The samples for EO analyses were collected over two growing seasons, representing two different experiments. For DNA analyses, only plant leaves were collected, while root and rhizome plants were collected for phytochemical and antioxidant testing. For the embryology analyses, flower buds and open flowers were collected in June, and seeds were collected in August-September when they reached maturity. Herbarium specimens of the *C. kellereri* from the two natural populations and *C. ruber* from Institute of Roses, Essential Oil and Medical Crops, in Kazanlak were deposited in the herbarium of the Agricultural University, Plovdiv, Bulgaria (herbarium numbers of the deposited *Centranthus* samples are 062637, 062638, 0622578, and 062636).

### 2.2. Methods

**2.2.1. Preparation of C. kellereri and C. ruber samples for the extraction of the essential oil (EO).** The samples of the two species were collected in June and were placed in a laboratory at a temperature of below 35˚C for air-drying.

**2.2.2 Essential oil (EO) extraction of C. kellereri and C. ruber.** The EO was extracted via 3-h hydro distillation of 100 g Centranthus aboveground biomass or roots using

Clevenger-type glass apparatus. The obtained EO was dried with anhydrous sodium sulfate, and placed in dark vials at 4˚C until gas chromatographic (GC) analysis

**2.2.3. Gas Chromatography Mass Spectrometry Flame Ionization Detection (GC-MS-FID) of essential oil (EO).** Oil samples in two replicates were studied by GC-FID and GC-MS as described earlier [23], with slight modifications. Briefly, GC analysis was performed on an Agilent 7890A system equipped with a FID and a DB-5 capillary column, 30 m × 0.25 mm, 0.25 μm, temperature programmed as follows: 40–300˚C at 5˚C/min. The carrier gas was nitrogen at a flow of 1.0 ml/min; the injector port and the detector temperatures were 230˚C and 280˚C, respectively. Samples were injected by splitting (split ratio 30:1).

GC/MS analysis was performed on an Agilent 7890A/5975C system with a DB-5 capillary column (30 m × 0.25 mm; 0.25 μm), as the operating conditions were the same as described above. The carrier gas was He (flow 1.0 ml/min). Mass spectra were taken at 70 eV in electron impact (EI) mode and the scan mass range was from 40–400 m/z. The ionization source, the transfer line, and the injector temperatures were set at 230˚C, 280˚C, and 250˚C, respectively.

The quantitative data were obtained from the electronic integration of the FID peak areas. The components of the oil samples were identified by their retention time, retention indices, relative to $C_8$-$C_{40}$ n-alkanes, matching with the Adams [24] and NIST'08 [25] libraries and by comparison of their mass spectra with data already available in the literature. The percentage of the identified compounds was computed from the GC peaks areas without any correction factors.

**2.2.4. Phytochemical analyses.** Dried above ground plant parts and roots were extracted and analyzed essentially as described by Zheljazkov et al. [26]. Extract constituents were characterized by retention time, accurate mass, isotopic pattern, and MS/MS spectra and identified by LC-MS/MS comparison with 500 authentic standards (Natural Products Library, Enzo Life Sciences). The distribution and relative abundance of the phytochemicals was visualized by the open-access software tool MetaboAnalyst 5.0 [27].

**2.2.5. DNA extraction.** Total genomic DNA was isolated from silica-dried leaves using standard CTAB extraction procedure [28]. Quantitative and qualitative tests of the DNA were done using a Nanodrop ND-1000 spectrophotometer. DNA samples were diluted to 25ng/μL, and stored at -20˚C before amplification.

*ISSR amplification product analysis.* Firstly, the polymorphism of 30 markers was tested and then seven polymorphic and reproducible ISSR primers (Microsynth, Balgach, Switzerland) were selected (Table 1). Polymerase chain reactions (PCR) were performed according to Petrova et al. [29]. PCR reactions were carried out in a Techne TC-5000 gradient thermal circler (Techne, Staffordshire, UK). The reproducibility of the technique was tested by replicating each amplification reaction twice. Amplification products were separated on 2% agarose gels stained by incorporating 1% GelRed (Biotium Inc., USA) at 1.5h, 135V along with 100 bp Plus

**Table 1. ISSR primers used for analysis of genetic diversity of different *Centranthus* species in Bulgaria.**

| Primer | Sequence 5'- 3' | Total number of bands | Number of polymorphic bands | Polymorphism (%) | Annealing temperature (˚C) |
|---|---|---|---|---|---|
| ISSR1 | $(AG)_{10}C$ | 10 | 7 | 70 | 60 |
| ISSR2 | $(AG)_8YT$ | 9 | 9 | 100 | 60 |
| ISSR3 | $(AC)_8G$ | 10 | 8 | 80 | 55 |
| ISSR4 | $(AC)_8YG$ | 8 | 6 | 75 | 60 |
| ISSR5 | $(AC)_8YT$ | 12 | 10 | 83 | 60 |
| ISSR6 | $(AC)_8YG$ | 9 | 7 | 78 | 55 |
| ISSR7 | $(AC)_8YA$ | 11 | 7 | 64 | 60 |
| Total | | 69 | 54 | 79 | |

DNA Ladder (Thermoscientific, Vilnius, Lithuania). The DNA fragments were visualized under UV light and further analyzed with a video image analyzer (BioImaging Systems, Cambridge, UK).

*Data analysis.* ISSR bands (clear and unambiguous) were scored using the binary system as present (1) versus absent (0). To learn about the genetic diversity among populations, we calculated the percentage of polymorphic loci (*P*), genetic diversity (*h*), unbiased genetic diversity and Shannon's information index (*SI*) in the software GenAlEx 6.5 [30]. The partitioning of genetic variability among populations, and Hierarchical Analysis of Molecular Variance (AMOVA) was performed with GenAlEx 6.5 by resembling 999 times. Genetic differentiation among populations ($F_{PT}$) based on the Euclidean distances was calculated with GenAlEx 6.5 and the latter were used to carry out a Principal Coordinate Analysis (PCoA). To investigate the genetic structure and the degree of admixture between each sample and between the three *Centranthus* populations, the Bayesian clustering method of STRUCTURE 2.3.4 [31] was used.

The simulations were conducted with a burn-in of 500,000 generations followed by Markov Chain Monte Carlo (MCMC) of 1,000,000 iterations. The *K* value was set to run from 1 to 10, each with 10 replications for reliability. The optimal number of clusters was estimated using the Evanno method [32] in Structure Harvester [33].

**2.2.6. Embryological analyses.** To estimate the reproductive capacity of the two target *Centranthus* species embryological studies (revealing the features of male and female gametophytes and the processes running in the reproductive structures) and an evaluation of basic parameters of the reproductive potential (pollen and seed viability) were carried out. For this purpose, the material (flowers and buds at different stages of development), collected from two populations of *C. kellereri* and a cultivated *C. ruber* described in Section 2.1 were fixed in FAA [34]. Subsequent processing was carried out according to the classical paraffin methods [34]. Cuttings on the initial material with a thickness of 8 to 18 μm (depending on the developmental stage) were made with a rotary microtome "Leica". The staining was performed with hematoxylin [35]. Permanent slides were prepared by including the cuts in Enthelan. The observations on the permanent slides were performed using light microscope Olympus CX21 (Olympus Corporation, Tokyo, Japan). The photographs were made with "Infinity lite" digital camera, 1.4 Mpx.

*Pollen viability.* For the estimation of pollen viability, the mature pollen grains in 30 anthers of the populations of the two *Centranthus* species were counted (on visible field at magnification 100x or 400x depending on their size) [35]. The count was performed on the permanent slides described above: the pollen grains as viable were estimated the ones with a distinguishable vegetative and generative cell with clear sculpture, and as non-viable—the degenerate dark-colored, deformed pollen grains (i.e., sterile, non-functional).

*Seed viability testing.* A rapid 24-hour test known as the tetrazolium test [36] was used to evaluate seed viability. For this purpose, embryos isolated from mature seeds pre-soaked in water at 30–35˚C were incubated in a diluted 1% solution of 2,3,5-triphenyltetrazolium chloride for 24 hours at 25˚C. The initially colorless tetrazolium solution turned red upon contact with the hydrogen from the respiratory enzymes in the seeds (embryos) [36]. Embryos with active respiratory activity were colored red by the tetrazolium solution and were considered viable. The more intensive the coloration, the more active was the respiratory activity of the embryo, while pink-colored embryos indicated lower viability than those colored in dark red. Colorless embryos were considered non-viable. The assessment was made according to the criteria for interpretation of tetrazolium test results [37], based on the color intensity and localization of the colored and uncolored parts after testing: embryos stained in dark red and pink and partially stained embryos (only the root tip colored in red) were considered viable, while colorless embryos or partially stained ones (only the top of the embryo–cotyledons are colored

in red) were considered non-viable [37]. Observations and micrographs were taken using a light microscope Olympus CX21 and an "Infinity lite" digital camera with 1.4 Mpx.

**2.2.7. Scanning electron microscopy (SEM) analysis.** The scanning electron microscope (SEM) analysis was performed using an FEI Quanta 600 SEM at the Oregon State University Microscopy Facility. The samples for SEM analysis were prepared by immersing subsamples into a fixative containing 1% paraformaldehyde and 2.5% glutaraldehyde in 0.1M sodium cacodylate buffer with a pH of 7.4. The fixation procedure continued for 2 hours, followed by two rinses in 0.1M cacodylate buffer for 15 minutes each. Afterwards, the samples were subjected to a dehydration series in acetone as described previously [26]. After dehydration, the samples were left to vent for 5 minutes, and the same procedure was repeated. The dry samples were then mounted onto an aluminum SEM stub with double-stick carbon tape. Samples on the sample holders were sputter-coated with a Cressington 108A sputter coater from Ted Pella with Au/Pd, 60/40 mix. The description of cells, shape, as well as the structure of the surfaces of leaves, stem, and seeds were determined according to Barthlott et al. [38].

**2.2.8. Analysis of total phenols, total flavonoids and antioxidant activity.** The roots of *C. kellereri* and *C. ruber* were studied. The locations of two species were described in the Section 2.1

*Plant material and extraction method.* One gram of ground and dried roots and 10 mL of solvent were used to extract phenolic and flavonoid compounds. Each variant was placed in a capped tube for the duration of the extraction. Three different solvents were used: 70% acidic methanol, 70% ethanol, and distilled water. Each variant was performed in 4 replicates. The ratio of dry plant material to extractant was 1:10 (1 g to 10 mL). The extraction lasted for 24 hours at room temperature and then for 6 days at 4˚C in the refrigerator. After extraction, the samples were filtered, and the resulting filtrates were used for analysis as described previously [26].

*Analysis of total phenols.* Quantitative determination of total phenols was performed by the method of Singleton et al. [39] with minor modifications [40]. A 40 μL sample (filtrate prepared as described above), 3160 μL of distilled water, 200 μL of Folin-Ciocalteu reagent and 300 μL of 20% (w/v) $Na_2CO_3$ were mixed in a tube. The tubes were incubated for 30 minutes at 75˚C and after cooling, the absorbance was recorded spectrophotometrically at 760 nm. The quantity of total phenols was calculated from a standard gallic acid curve with the formula $\hat{y} = 1.421x + 0.0074$ ($R^2 = 0.9997$) and was expressed as mg of gallic acid equivalents per granular dry weight [39].

*Analysis of total flavonoids.* Total flavonoids were determined using the method described by Zhishen et al. [41]. For this, 1 mL of sample (filtrate), 4 mL of distilled water, and 300 μL of 5% (v/v) $NaNO_2$ were mixed. After 5 minutes, 300 μL of 10% $AlCl_3$ was added, and after an additional 6 minutes, 2 mL of 1M NaOH was added. The volume was immediately made up to 10 mL with distilled water. The absorbance was measured spectrophotometrically at a wavelength of 510 nm. The total amount of flavonoids was calculated from a standard quercetin curve with the formula $\hat{y} = 0.0776x + 0.0244$ ($R^2 = 0.9963$) and was expressed as mg of quercetin equivalents per gram of dry weight [41].

*DPPH assay.* To prepare the DPPH reagent, 0.012 g was dissolved in 100 mL of absolute ethanol and placed in an ultrasonic bath two times for 15 min. This solution was used to determine the antiradical activity of plant extracts by the method of Brand-Williams et al. [42] with minor changes [40]. For this purpose, 1.8 mL of absolute ethanol, 0.6 mL of DPPH solution, and 0.6 mL of sample were mixed in the reaction tubes. Absorption was recorded after 10 minutes at 517 nm against absolute alcohol. The DPPH radical scavenging activity of the sample was expressed as mg Trolox equivalent antioxidant capacity (TEAC) by formula obtained from standard curve ($\hat{y} = -0.1954x + 0.708$, $R^2 = 0.9858$) [42].

**Table 2. 2018 *C. kellereri* species: ANOVA *p*-values that show the significance of the effects of location and plant part nested in location [plant part (location)] on 8 compounds.** Significant effects that require multiple means comparison are shown in bold.

| Compound | Source of variation | |
|---|---|---|
| | **Location** | **Plant part (Location)** |
| Furfural | 0.001 | **0.001** |
| 3-Methylbutanoic acid (isovaleric acid) | 0.308 | **0.001** |
| 3-Methylpentanoic acid (3-methylvaleric acid) | 0.002 | **0.001** |
| Benzene acetaldehyde (hyacinthin) | 0.216 | **0.001** |
| Borneol | 0.256 | **0.001** |
| Others (aldechyde, alcohol, ethers) | 0.760 | **0.001** |
| Monoterpenoid | 0.008 | **0.001** |
| Methyl-branched fatty acid | 0.013 | **0.001** |

*ABTS assay.* The ABTS assay included the preparation of stock and working solutions following an approved procedure. The method was a modification of Xiao et al. [43]. The stock solution of ABTS (7 mM) and $K_2S_2O_8$ (140 mM) were kept in dark (0–4°C) [43] and were used for working solution preparation. Then, the activated ABTS radical-containing solution was diluted with distilled water to the final absorbance of 0.700±0.05 at 734 nm [43].

*ABTS assay procedure.* A 100 mL sample (or Trolox standard) and 2.900 mL ABTS working solution were mixed in tubes. The tubes were incubated for 5 min and then the absorbance was measured at 734 nm wavelength [43]. The Trolox was used as a standard and distilled water as the blank control. The ABTS radical scavenging activity of sample was expressed as mg Trolox equivalent antioxidant capacity (TEAC) by formula obtained from standard curve ($\hat{y}$ = -0.0446x + 0.8223, $R^2$ = 0.984) [43].

**2.2.9. Statistical analyses of EO samples and the ANOVA results.** *Samples from the 2018 growth season experiment.* For *C. kellereri* species, the effects of (1) Location (population) (2 levels: Bansko and Vratsa), and (2) Plant part within each Location (2 levels: Aboveground and Roots) on 8 compounds (furfural, 3-methylbutanoic acid (isovaleric acid), 3-methylpentanoic acid (3-methylvaleric acid), benzene acetaldehyde (hyacinthin), borneol, others (aldechyde, alcohol, ethers), monoterpenoid, and methyl-branched fatty acid) was determined using a nested design where Plant part is nested in Location. The components of the model are shown in the Source of Variation row of Table 2. The multiple means comparison results are shown in Table 3.

For *C. ruber* species, where the data were collected only from Kazanlk location and from Roots Plant part, only descriptive statistics (average of the two replications) are reported in Table 4.

*Samples from the 2019 growing season experiment.* In *C. kellereri* species, the effects of (1) Location (population) (2 levels: Bansko, and Vratsa), and (2) Plant part within each Location (2

**Table 3. 2018 growing season *C. kellereri* species: Mean concentration (%) of furfural, 3-methylbutanoic acid (isovaleric acid) (Methy-I), 3-methylpentanoic acid (3-methylvaleric acid) (Methy-3-M), benzene acetaldehyde (hyacinthin) (benzene), borneol, others (aldechyde, alcohol, ethers) (Other), monoterpenoid (Monoter), and methyl-branched fatty acid (Methyl-b) obtained from the four combinations of Location and Plant part.**

| Plant part (Location) | Furfural | Methy-I | Methy-3-M | Benzene | Borneol | Other | Monoter | Methyl-b |
|---|---|---|---|---|---|---|---|---|
| Aboveground (Bansko) | 4.9 a* | 62.4c | 6.22c | 3.79a | 14.33a | 11.59a | 18.26a | 68.67c |
| Aboveground (Vratsa) | 1.02 b | 71.7 a | 14.7 b | 0.00 c | 0.13 b | 6.33 c | 4.96 b | 86.45b |
| Roots (Bansko) | 5.25 a | 67.5 b | 20.61a | 0.61 b | 0.00 b | 7.23 b | 2.36 c | 88.08a |
| Roots (Vratsa) | 4.86 a | 60.16 c | 6.64 c | 4.05 a | 15.14 a | 12.34 a | 19.18 a | 66.80 d |

*Within each column, means sharing the same letter are not significantly different at the 5% level of significance.

**Table 4. 2018 *Centranthus ruber*, Kazanlk location, and roots plant part: Mean concentration (%) of the 8 compounds listed in the table.**

| Compound | Concentration (%) |
|---|---|
| Furfural | 5.31 |
| 3-Methylbutanoic acid (isovaleric acid) | 77.53 |
| 3-Methylpentanoic acid (3-methylvaleric acid) | 10.08 |
| Benzene acetaldehyde (hyacinthin) | 0.58 |
| Borneol | 0.00 |
| Others (aldechydes, alcohol, ethers) | 7.22 |
| Monoterpenoid | 3.33 |
| Methyl-branched fatty acid | 87.61 |

levels: Root and Stem) on 18 compounds (3-methylbutanoic acid (isovaleric acid), 3-methylpentanoic acid (3-methylvaleric acid), germacrene D, n-tetradecanal, geranyl valerate, (2Z,6E)-farnesyl acetate, geranyl linalool, (9E,12E,15E)-octadecatrien-1-ol, n-octadecanol, (9E,12E,15E)-octadecatrienal, 3,7,11,15-tetramethyl-2-hexadecen-1-ol, (Z,Z)-9,12-octadecadienoic acid, (Z,Z,Z)-9,12, 15-octadecatrienoic acid, monoterpenes, monoterpenoids, sesquiterpenes, sesquiterpenoids, long-chain alkane, fatty acid (esters, long-chain, methyl-branched), and others (esters, alcohol, fatty aldehyde)) was determined using a nested design where Plant part is nested in Location. The components of the model are shown in the Source of Variation row of Table 5. The multiple means comparison results are shown in Tables 6 and 7.

*C. ruber*. Since there is no enough error degree of freedom to do a test of hypothesis to test if there is a significant difference between the Aboveground and Roots, only descriptive statistics (averages of the 2 replications) are shown below (Table 8).

**Table 5. 2019 Experiment.** *C. kellereri* species: ANOVA *p*-values that show the significance of the effects of Location and Plant part nested in Location on 18 compounds. Significant effects that require multiple means comparison are shown in bold.

| Compound | Source of variation | |
|---|---|---|
| | Location | Plant part (Location) |
| 3-Methylbutanoic acid (isovaleric acid) | 0.001 | **0.001** |
| 3-Methylpentanoic acid (3-methylvaleric acid) | 0.001 | **0.001** |
| Germacrene D | 0.001 | **0.004** |
| n-Tetradecanal | 0.001 | **0.002** |
| Geranyl valerate | 0.001 | **0.001** |
| (2Z,6E)-Farnesyl acetate | 0.001 | **0.001** |
| Geranyl linalool | 0.149 | **0.001** |
| (9E,12E,15E)-Octadecatrien-1-ol | 0.001 | **0.001** |
| n-Octadecanol | 0.001 | **0.001** |
| (9E,12E,15E)-Octadecatrienal | 0.001 | **0.001** |
| 3,7,11,15-Tetramethyl-2-hexadecen-1-ol | 0.001 | **0.001** |
| (Z,Z)-9,12-octadecadienoic acid | 0.002 | **0.001** |
| (Z,Z,Z)-9,12, 15-octadecatrienoic acid | 0.001 | **0.001** |
| Monoterpenes (monoterpenoids) | 0.001 | **0.001** |
| Sesquiterpenes (sesquiterpenoids) | 0.001 | **0.001** |
| Long-chain alkane | 0.001 | **0.001** |
| Fatty acid (esters, long-chain, methyl-branched) | 0.001 | **0.001** |
| Others (esters, alcohol, fatty aldehyde) | 0.001 | **0.001** |

**Table 6. 2019 Experiment.** *C. kellereri* species: Mean concentration (%) of 3-methylbutanoic acid (isovaleric acid) (3-Meth-Iso), 3-methylpentanoic acid (3-methylvaleric acid) (3-Meth-3-M), germacrene D (Ger-D), n-tetradecanal (n-Tet), geranyl valerate (Ger-val), (2Z,6E)-farnesyl acetate (2Z,6E), geranyl linalool (GL), (9E,12E,15E)-octadecatrien-1-ol (Oct-1), and n-octadecanol (n-Oct) obtained from the four combinations of Location and Plant part.

| Plant part (Location) | 3-Meth-Iso | 3-Meth-3-M | Ger-D | n-Tet | Ger-val | 2Z,6E | GL | Oct-1 | n-Oct |
|---|---|---|---|---|---|---|---|---|---|
| **Root (Bansko)** | 2.81 c* | 0.71 c | 2.44 b | 15.93 b | 3.33 a | 3.13 b | 4.38 b | 9.25 b | 1.80 c |
| **Root (Vratsa)** | 28.05 a | 6.69 b | 0.74 c | 12.34 c | 2.10 c | 3.50 b | 6.15 a | 5.78 c | 5.14 a |
| **Stem (Bansko)** | 0.56 d | 0.21 c | 3.36 a | 18.77 a | 1.05 d | 2.39 c | 3.67 c | 23.85 a | 3.47 b |
| **Stem (Vratsa)** | 4.32 b | 22.05 a | 0.33 d | 9.96 d | 2.95 b | 8.00 a | 1.52 d | 2.80 d | 1.56 c |

*Within each column, means sharing the same letter are not significantly different at the 5% level of significance.

Since only the *C. kellereri* species, Vratsa location and *C. ruber* and the two Plant parts (leaves and stems) have replications, it is not possible to do ANOVA. Instead, descriptive statistics (averages for these two Plant parts, and the actual observed values) are reported in Tables 9 and 10.

The analysis was conducted using the GLM procedure [44]. For each response variable, the validity of model assumptions on the error terms was verified by examining the residuals as described by Montgomery [45]. When normal distribution assumption is violated, an appropriate transformation was applied; however, their means reported in the tables are back-transformed to the original scale. Since Plant part nested in Location effect was significant ($p < 0.05$) in all compounds, multiple means comparison was conducted using the Fisher's LSD method at the 5% level of significance to compare all combinations of location and plant part. Even if the effect of Location is significant, if the effect of Plant part (Location) is significant, multiple means comparison is done only on Plant part (Location) because the differences among the plant parts vary with location.

**2.2.10. Antioxidant activity statistical analyses and results.** A 3 x 3 factorial design with 4 replications was used to determine the main and interaction effects of Population (Kazanlak, Bansko, and Vratsa) and Extraction method (EtOH70%, MeOH70%+HCL, and Water) on flavonoids, polyphenols, and two antioxidant activities (ABTS and DPPH). The analyses were conducted using the Mixed Procedure of SAS [44]. Since the interaction effect of population and extraction method was highly significant (*p*-value < 0.01) for all four response variables (Table 11), multiple means comparison was done to compare the nine combinations of population and extraction method using the lsmeans statement at the 5% level of significance (Table 12). For all four response variables, normal distribution and constant variance assumptions on the error terms were verified by examining the residuals as described in Montgomery [45].

**Table 7. 2019 Experiment.** *C. kellereri* species: Mean concentration (%) of (9E,12E,15E)-octadecatrienal (Octad), 3,7,11,15-tetramethyl-2-hexadecen-1-ol (3,7,11,15), (Z,Z)-9,12-octadecadienoic acid (Z,Z), (Z,Z,Z)-9,12, 15-octadecatrienoic acid (Z,Z,Z), monoterpenes (monoterpenoids) (Mono), sesquiterpenes (sesquiterpenoids) (Sesq), long-chain alkane (Long), fatty acid (esters, long-chain, methyl-branched) (Fatty), and others (esters, alcohol, fatty aldehyde) (Others) obtained from the four combinations of Location and Plant part.

| Plant part (Location) | Octad | 3,7,11,15 | Z,Z | Z,Z,Z | Mono | Sesq | Long | Fatty | Others |
|---|---|---|---|---|---|---|---|---|---|
| Root (Bansko) | 11.2 b* | 7.7 a | 4.10 b | 3.6 b | 3.40 b | 15.8 a | 3.5 b | 14.7 c | 60.9 b |
| Root (Vratsa) | 6.97 c | 1.1 d | 0.98 c | 1.1 c | 3.10 b | 6.9 c | 2.8 bc | 39.9 a | 45.8 d |
| Stem (Bansko) | 5.44 d | 4.2 b | 3.80 b | 5.5 a | 1.12 c | 11.1 b | 9.5 a | 11.9 d | 64.7 a |
| Stem (Vratsa) | 20.2 a | 2.1 c | 8.12 a | 0.5 d | 5.04 a | 4.06 d | 1.3 c | 36.7 b | 51.3 c |

*Within each column, means sharing the same letter are not significantly different at the 5% level of significance.

**Table 8. 2019 Experiment.** *C. ruber* species (Kazanlk): Mean concentration (%) of the compounds measured in the aboveground and roots.

| Compound name | Aboveground | Root |
|---|---|---|
| 3-Methylbutanoic acid (isovaleric acid) | 0.50 | 8.68 |
| 3-Methylpentanoic acid (3-methylvaleric acid) | 0.20 | 2.09 |
| Germacrene D | 4.01 | 0.48 |
| n-Tetradecanal | 20.5 | 6.04 |
| Geranyl valerate | 0.93 | 1.30 |
| (2Z,6E)-Farnesyl acetate | 3.15 | 7.68 |
| Geranyl linalool | 4.29 | 3.77 |
| (9E,12E,15E)-Octadecatrien-1-ol | 6.58 | 9.26 |
| n-Octadecanol | 3.08 | 4.71 |
| (9E,12E,15E)-Octadecatrienal | 22.51 | 18.73 |
| 3,7,11,15-Tetramethyl-2-hexadecen-1-ol | 3.95 | 3.24 |
| (Z,Z)-9,12-octadecadienoic acid | 3.40 | 5.88 |
| (Z,Z,Z)-9,12, 15-octadecatrienoic acid | 4.90 | 6.06 |
| Monoterpenes (monoterpenoids) | 1.00 | 0.57 |
| Sesquiterpenes (sesquiterpenoids) | 10.87 | 9.93 |
| Long-chain alkane | 8.47 | 4.89 |
| Fatty acid (esters, long-chain, methyl-branched) | 10.52 | 25.25 |
| Others (esters, alcohol, fatty aldehyde) | 67.31 | 57.89 |
| Total | 98.16 | 98.52 |

# 3. Results

## 3.1. Population genetic diversity study

**Population genetic diversity.** The 7 ISSR primers used generated 69 polymorphic bands, ranging in size between 150 and 3000 bp (Table 1). The primer $(AC)_8YT$ produced the highest number of bands (12) whereas the primer $(AC)_8YG$ produced the lowest number of bands (8), (Table 1). The percentage of polymorphic loci (*P*) varied from 66.67% (*C. kellereri*, Bansko) to 75.00% (*C. kellereri*, Vratsa). The *C. kellereri* Vratsa population exhibited the highest values of genetic indexes *h*, $h_u$ and *SI* of 0.255, 0.261 and 0.383, respectively (Table 13).

**Population genetic structure and differentiation.** Principal coordinates analysis (PCoA) was carried out to provide spatial representation of the genetic structure of the three

**Table 9. Chemical profile of the hexane-extracted samples:** Mean concentration (%) of 3-methylbutanoic acid (isovaleric acid) (3-Meth-Iso), 3-methylpentanoic acid (3-methylvaleric acid) (3-Meth-3-M), (2E)-tridecen-1-al (2E), n-tetradecanal (n-Tetrad), (2E,6E)-farnesyl acetate (2E,6E), n-hexadecanoic acid (n-Hex), and (Z,Z)-9,12-octadecadienoic acid (Z,Z).

| Species | Location | Plantpart | 3-Meth-Iso | 3-Meth-3-M | 2E | n-Tetrad | 2E,6E | n-Hex | Z,Z |
|---|---|---|---|---|---|---|---|---|---|
| *C. kellereri* | Vratsa | leaves | 3.81 | 1.20 | 4.26 | 41.7 | 7.29 | 4.79 | 20.2 |
| *C. kellereri* | Vratsa | stems | 2.37 | 0.60 | 3.14 | 36.0 | 2.01 | 5.09 | 17.9 |
| *C. kellereri* | Vratsa | roots | 65.9 | 10.70 | 1.47 | 0.60 | 0.20 | 6.40 | 0.44 |
| *C. kellereri* | Bansko | leaves | 3.28 | 1.41 | 2.45 | 33.8 | 2.33 | 4.99 | 42.1 |
| *C. kellereri* | Bansko | stems | 7.83 | 0.34 | 1.49 | 51.0 | 2.04 | 3.60 | 24.1 |
| *C. kellereri* | Bansko | roots | 31.0 | 8.18 | 2.20 | 2.08 | 0.41 | 4.57 | 0.71 |
| *C. ruber* | Kazanlk | roots | 33.8 | 5.95 | 0.94 | 6.14 | 0.50 | 5.29 | 0.86 |
| *C. ruber* | Kazanlk | leaves | 4.37 | 1.24 | 0.23 | 36.4 | 3.29 | 14.2 | 22.0 |
| *C. ruber* | Kazanlk | stems | 6.96 | 0.88 | 0.93 | 44.8 | 4.17 | 9.68 | 18.3 |
| *C. ruber* | Kazanlk | Flowers | 2.04 | 0.55 | 0.26 | 0.22 | 2.06 | 1.86 | 13.8 |

**Table 10. Chemical profile of the hexane-extracted samples: Mean concentration (%) of n-tetracosane (n-Tetrac), others (aldechyde, alchohol, ethers) (Other), monoterpenoid (Mono), sesquiterpenoids (Sesq), fatty acid (Fatty), and long-chain alkane (Long).**

| Species | Location | Plantpart | n-Tetrac | Other | Mono | Sesq | Fatty | Long |
|---|---|---|---|---|---|---|---|---|
| *C. kellereri* | Vratsa | Leaves | 4.61 | 46.43 | 7.29 | 1.00 | 37.23 | 6.52 |
| *C. kellereri* | Vratsa | Stems | 25.71 | 40.46 | 2.01 | 0.09 | 28.29 | 27.46 |
| *C. kellereri* | Vratsa | Roots | nd* | 2.67 | 2.23 | 5.00 | 85.36 | 0.00 |
| *C. kellereri* | Bansko | Leaves | 3.61 | 36.42 | 2.33 | 0.81 | 52.28 | 6.18 |
| *C. kellereri* | Bansko | Stems | 3.18 | 52.98 | 2.04 | 0.74 | 36.63 | 5.15 |
| *C. kellereri* | Bansko | Roots | nd | 5.26 | 28.68 | 17.00 | 47.27 | 0.00 |
| *C. ruber* | Kazanlk | Roots | nd | 8.06 | 3.57 | 37.55 | 49.56 | 0.00 |
| *C. ruber* | Kazanlk | Leaves | 9.34 | 36.66 | 3.29 | 2.90 | 44.74 | 9.87 |
| *C. ruber* | Kazanlk | Stems | 7.32 | 45.92 | 4.17 | 0.42 | 36.83 | 10.36 |
| *C. ruber* | Kazanlk | Flowers | 10.50 | 0.48 | 2.36 | 0.38 | 44.17 | 49.99 |

*nd = not detected.

*Centranthus* populations. The first two principal coordinates accounted for 46.93% of the total variance (28.76% and 18.17%, respectively), Fig 1. The results showed a clear differentiation between species; populations of different *Centranthus* localities were genetically distant and the individuals form three major groups, only few *C. kellereri* individuals fell into the *C. ruber* population cluster. Population genetic structure was further studied using a Bayesian clustering algorithm; perform in STRUCTURE (Fig 2). According to the method described by Evanno et al. [32], the investigated *Centranthus* individuals were optimally divided into three main clusters (Fig 2A and 2B). The result of Analysis of molecular variance (AMOVA) showed that 66% of the variation to reside within populations, and 34% among investigated populations. A significant genetic difference was found among investigated populations ($F_{PT}$ = 0.341, $P$ < 0.001) (Table 14).

## 3.2. Essential oils analysis of *C. kellereri* and *C. ruber*

**3.2.1. Qualitative composition of the essential oil (EO).** Statistical analyses of the 2018 data indicated that the ANOVA *p*-values for plant part were significant for the EO compounds (Table 2).

*Composition of C. kellereri EO in the 2018 experiment.* In the 2018 experiment, the concentration of furfural in the *C. kellereri* EO from aboveground plants parts was higher in the samples from Bansko (4.9%) and lower in the samples from Vratsa (1.02%). However, furfural concentration in the roots EO was not significantly different between the plants at the two locations (populations) (5.25 and 4.86%, at Bansko and Vratsa, respectively) (Table 3).

Methylvaleric acid (isovaleric acid) was the major constituent of the EO in *C. kellereri*, ranging between 60.2% and 71.7% of the total EO, with the highest concentrations in the aboveground plant parts from Vratsa and the lowest in the roots from Vratsa. 3-Methylvaleric acid

**Table 11. *p*-values of the ANOVA that show the significance of the main and interaction effects of population and extraction method.** Significant effects that require multiple means comparison are shown in bold.

| Source of variation | Flavonoids | Polyphenols | ABTS | DPPH |
|---|---|---|---|---|
| Population | 0.001 | 0.028 | 0.001 | 0.001 |
| Extraction Method | 0.001 | 0.001 | 0.001 | 0.005 |
| Population*Extraction Method | **0.006** | **0.001** | **0.001** | **0.001** |

**Table 12. Mean flavonoids (mg QE/g DW), polyphenols (mg GAE/g DW), ABTS (mg TE/g DW), and DPPH (mg TE/g DW) obtained from the nine combinations of population and extraction method.** Within each column, means sharing the same letter are not significantly different.

| Species | Population | Extraction method | Flavonoids | Polyphenols | ABTS | DPPH |
|---|---|---|---|---|---|---|
| *C. ruber* | Kazanlak | EtOH70% | 226.2 b | 7.98 a | 1.81 a | 0.11 c |
| *C. ruber* | Kazanlak | MeOH70%+HCL | 81.3 e | 0.59 d | 0.67 d | 0.13 bc |
| *C. ruber* | Kazanlak | Water | 128.7 d | 3.23 c | 1.82 a | 0.15 bc |
| *C. kellererii* | Bansko | EtOH70% | 220.7 b | 6.38 b | 1.78 a | 0.23 a |
| *C. kellereri* | Bansko | MeOH70%+HCL | 88.8 e | 0.77 d | 0.88 c | 0.15 b |
| *C. kellereri* | Bansko | Water | 155.2 c | 3.64 c | 1.81 a | 0.24 a |
| *C. kellereri* | Vratsa | EtOH70% | 302.9 a | 6.33 b | 1.76 a | 0.26 a |
| *C. kellereri* | Vratsa | MeOH70%+HCL | 156.2 c | 0.83 d | 1.09 b | 0.25 a |
| *C. kellereri* | Vratsa | Water | 208.6 b | 3.70 c | 1.81 a | 0.23 a |

was another major constituent in the EO, ranging from 6.2% (in the biomass EO from Bansko) to 20.6% in the roots EOs from Bansko, with significant differences between the locations (Table 2). Overall, the concentration of 3-methylvaleric acid in the EO of *C. kellereri* was higher in the aboveground plant parts than in the roots in Vratsa. However, the reverse trend was observed in the plants from Bansko, which had a higher concentration of this compound in the roots. The concentration of benzene acetaldehyde in the EO varied from undetected (in the aboveground plant parts from Vratsa) to 4.1% in the roots from the same location. Again, the reverse trend was observed in the plants collected at Bansko, which had a more abundant concentration of this compound in the aboveground plant parts than in the roots (Table 3). The concentration of borneol also varied depending on the location and the plant part, ranging from undetected amounts in the roots from Bansko to 14.3% in the aboveground parts from Bansko and 15.1% in the roots from Vratsa. Other unidentified constituents in the EO varied from 6–7% to 11–12%. The concentration of monoterpenes was highest in the aboveground parts from Bansko and in the roots from Vratsa, ranging from 2.3 to 19.2% of the total oil (Table 3).

*Composition of C. ruber root EO in the 2018 experiment*. The concentration means of the 8 EO compounds from the roots of *C. ruber* are shown in Table 4. Generally, the concentration of *C. ruber* EO compounds from roots was similar to their respective concentrations in *C. kellereri* roots (Tables 4 and 5). The concentration of furfural in the root EO of the two species was similar; however, the concentration of 3-methylbutanoic acid was higher in *C. ruber* than in *C. kellereri*. Furthermore, borneol was not detected in *C. ruber* root EO, similar to the *C. kellereri* EO from Bansko, while benzene acetaldehyde and the monoterpenes concentrations in root EO of *C. ruber* and *C. kellereri* from Bansko were similar. Also, the concentration of 3-methylpentanoic acid in *C. ruber* was around 10%, which was between the concentrations of this compound in the *C. kellereri* root from the two locations, Bansko and Vratsa.

**Table 13. Genetic diversity indices of investigated *Centranthus* populations.**

| Population | P (%) | h | $h_u$ | SI | Private bands |
|---|---|---|---|---|---|
| *C. kellereri*, Bansko | 66.67 | 0.175 (0.037) | 0.179 (0.038) | 0.275 (0.053) | 0 |
| *C. kellereri*, Vratsa | 75.00 | 0.255 (0.040) | 0.261 (0.041) | 0.383 (0.055) | 3 |
| *C. ruber* | 70.83 | 0.238 (0.039) | 0.244 (0.040) | 0.360 (0.055) | 3 |
| Mean (SE) | 70.83 (2.41) | 0.222 (0.022) | 0.228 (0.023) | 0.339 (0.031) | 2 |

P: percentage of polymorphic loci; h: genetic diversity, $h_u$: unbiased genetic diversity; SI: Shannon's information index.

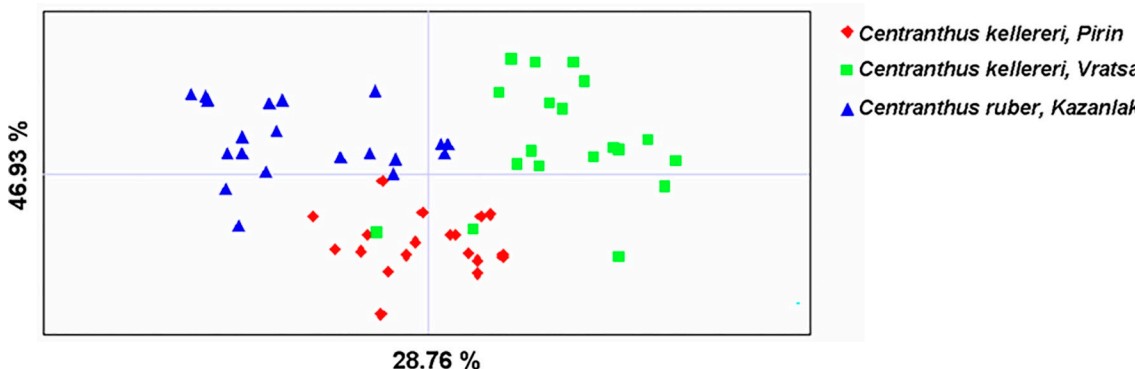

**Fig 1. A two-dimensional plot of the PCoA of all analyzed *Centranthus* individuals.**

*The 2019 Centranthus experiments.* As indicated, in 2019, separate experiments were conducted using *C. kellereri* samples from the two locations (populations), Bansko and Vratsa, as well as commercially grown *C. ruber* at the experimental fields of the Institute for Roses, Aromatic and Medicinal Plants in Kazanluk. The samples collected in the 2019 experiment were from slightly different interpopulation locations compared to those in 2018, and they were collected at different timeframes and growth stages, and underwent different sample processing, such as EO isolation. Therefore, the results from the two experiments conducted in 2018 and 2019 cannot be compared.

The ANOVA *p*-values that show the significance of the effects of Location and Plant part nested in Location on 18 compounds from the 2019 Experiments are shown in Table 5, and the multiple means comparison results are shown in Tables 6 and 7. The concentration of 3-methylbutanoic acid in the root and aboveground plant parts EO varied significantly, ranging from 0.56% to 28%. Generally, the concentration was the highest in the roots from Vratsa, followed by roots from Bansko, aboveground parts from Vratsa, and aboveground parts from Bansko, with significant differences between them. The concentration of 3-methylpentanoic

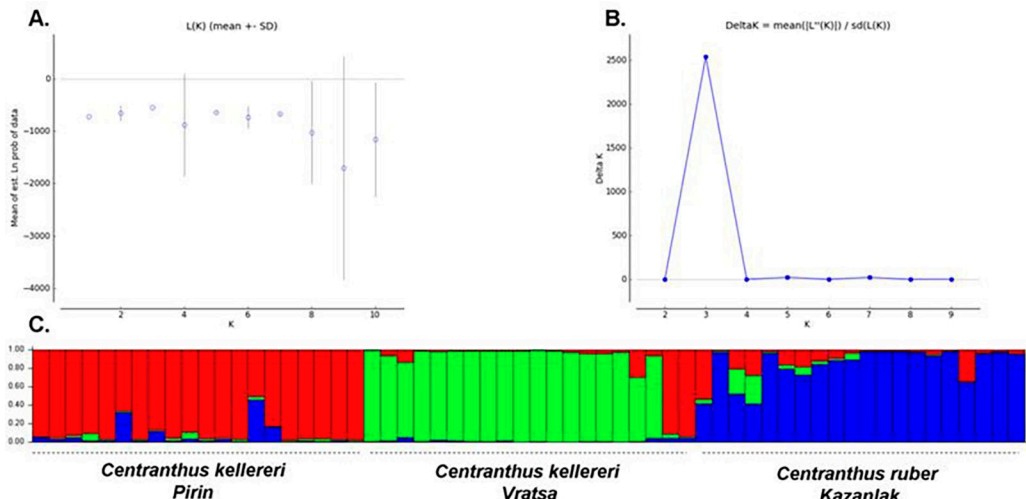

**Fig 2.** (A) Relationship between the number of determined groups (K) and LnP(D); (B) Relationship between the number of determined groups (K) and estimated ΔK; (C). Genetic relationships among the populations of *Centranthus* estimated using STRUCTURE program based on ISSR data. STRUCTURE clustering results for K = 3 to as suggested in (B).

**Table 14. Analysis of molecular variance based on the ISSR markers for the three populations of *Centranthus*.**

| Source of variation | df | SS | MS | Variance component | Percentage total (%) | F statistic | P |
|---|---|---|---|---|---|---|---|
| Among populations | 2 | 66.5 | 33.25 | 1.516 | 34 | | |
| Within populations | 57 | 166.9 | 2.93 | 2.928 | 66 | $F_{PT}$ = 0.341 | 0.001 |
| Total | 59 | 233.4 | | 4.444 | 100 | | |

acid in the EO also varied significantly, from 0.2 to 22% of the oil, in the following order: aboveground parts Vratsa > roots Vratsa > roots Bansko = aboveground parts Bansko (Table 6).

The concentration of germacrene D in the EO varied significantly from 0.33 to 2.44% in the following order: aboveground parts Bansko > roots Bansko > roots Vratsa > aboveground parts Vratsa. The concentration of n-tetradecanal in the EO also varied significantly from 9.9 to 18.8%, with the highest concentration of the aboveground parts from Bansko and the lowest in the aboveground parts of Vratsa. The concentration of geranyl valerate in the EO was 1–3% of the total, with the highest in roots from Bansko and lowest in the aboveground parts from Bansko. (2Z,6E)-Farnesyl acetate concentration of the EO varied from 2.4 (aboveground parts Bansko) to 8% (aboveground parts Vratsa) of the total oil. Geranyl linalool varied from 1.5% (aboveground Vratsa) to 6.2% (roots Vratsa). (9E,12E,15E)-Octadecatrien-1-ol (Oct-1) was one of the major EO constituents and varied from 2.8% (aboveground Vratsa) to 23.9% (aboveground Bansko). n-Octadecanol (n-Oct) concentration was between 1.6% (aboveground Vratsa) to 5.1 (roots Vratsa) (Table 6).

(9E,12E,15E)-Octadecatrienal concentration in *C. kellereri* varied from 5.4% (aboveground parts at Bansko) to 20% (aboveground parts at Vratsa) (Table 7). Tetramethyl-2-hexadecen-1-ol varied from 1.1% (roots at Vratsa) to 7.7% (roots at Bansko) of the total EO, while 9,12-octadecadienoic acid was from 0.98% (roots Vratsa) to 8.1% (aboveground Vratsa), and 9Z,12Z,15Z-octadecatrienoic acid was from 0.5% (aboveground Vratsa) to 5.5% (aboveground Bansko).

Overall, the monoterpenes constituted 1.1% (aboveground Bansko) to 5% (aboveground Vratsa), sesquiterpenes were 4% (aboveground parts Vratsa) to 15.8% (roots Bansko), and long-chain alkanes were 1.3% (aboveground parts Vratsa) to 9.5% (above ground parts at Bansko). Fatty acids constituted major part of the EO from 11.9% (above ground parts Bansko) to 39.9% (roots Vratsa) (Table 6). Other constituents (esters, alcohol, and fatty aldehyde) of the EO from the 2019 Experiments were the major part of the EO with an overall concentration from 45.8% (roots Vratsa) to 64.7% (aboveground Bansko) (Table 7).

*EO constituents of C. ruber from 2019.* The mean concentration (%) of the compounds measured in the aboveground and roots EO of *C. ruber* are provided in Table 8. Overall, 3-methylbutanoic acid, 3-methylpentanoic acid, and fatty acids (esters, long-chain, methyl-branched) were more abundant in root EO, while germacrene D, n-tetradecanal and long-chain alkanes were more abundant in the aboveground EO of *C. ruber* (Table 8). The *C. ruber* EO from roots and aboveground plant parts showed similar abundance of monoterpenes, monoterpenoids, sesquiterpenes and sesquiterpenoids (Table 8).

*Hexane extracted EO of C. kellereri and C. ruber.* The chemical profile of the hexane-extracted EOs from the *C. kellereri* and *C. ruber* samples is shown in Tables 9 and 10. Overall, 3-methylbutanoic acid was most abundant in the roots of *C. kellereri* from Vratsa (65.9%). Also, 3-methylpentanoic acid was the most abundant in *C. kellereri* roots from Vratsa and Bansko, (2E)-tridecen-1-al was most abundant in the leaves and stems from Vratsa. n-Tetradecanal was the most abundant in the *C. kellereri* from Bansko, leaves and stems of *C. kellereri* in Vratsa, leaves and stems in *C. ruber*. (2E,6E)-Farnesyl acetate (2E,6E), was the most abundant

in the leaves of *C. kellereri* from Vratsa, n-hexadecanoic acid, was the most abundant in the *C. ruber* leaves and stems, and (Z,Z)-9,12-octadecadienoic acid was the most abundant in the *C. kellereri* from Bansko (Table 9). Furthermore, n-tetracosane was the most abundant in the *C. kellereri* stems from Vratsa (25.7%) and not detected in *C. kellereri* and *C. ruber* roots.

Other compounds such as the sum of aldehydes, alcohols, ethers (Other), were most abundant in the stems and leaves of *C. kellereri* from the two locations and in *C. ruber*, with lesser amounts in the roots of the two species (Table 10). Also, monoterpenoids were the most abundant in the roots of *C. kellereri* from Bansko, while sesquiterpenoids were most abundant in the *C. kellereri* from Bansko and in roots of *C. ruber*. Fatty acids (Fatty) were major EO constituents in all EO, most abundant in the *C. kellereri* roots EO from Vratsa, while the long-chain alkanes were most abundant in the *C. ruber* flowers (Table 10).

### 3.3. Embryological research

The observation on the permanent slides revealed the following peculiarities of the male and female generative sphere of the two studied species:

*3.3.1. Male generative sphere.* The anther is tetrasporangiate (Fig 3A). The anther wall develops by the Dicotyledonous type according to Davis classification [46] and consists of four layers: epidermis, endothecium, one middle layer and tapetum that at the beginning of the anther ontogenesis do not differ from each other. During the meiosis in microspore mother cells (MMCs) and development of male gametophyte, the anthers layers undergo changes: the epidermal cells enlarged and rounded up outside becoming toothed, with thickened outer wall (Fig 3B and 3D); the endotecium developed fibrous thickenings after the stage of one-nucleate pollen and as result of division in tangential direction of some of them, the layer become partially two rowed (Fig 3B); the cells of middle layer become flattened, pressed from the growing cells of the endothecium and tapetum and the layer degenerated about the second (homotypic) division of meiosis in MMCs; the tapetum differentiated into atypical amoeboid one, atypical after the formation of unicellular pollen (Fig 3C). Two-layered sporogenous tissue and predominantly tetrahedral microspore tetrads formed after simultaneous microsporogenesis in MMCs were observed. The mature pollen grains are 3-celled (with generative cell and two sperm) (Fig 3B), 3-colpato-porate (Fig 3D).

**3.3.2. Female generative sphere.** The ovary is 3-locular, but only one of the three locules is fertile (Fig 4A). The ovule is anatropous, tenuinucellate and unitegmic (Fig 4B). The funiculus is massive, enough long, and its epidermal cells do not form obturator. Within the still hemitropous ovule unicellular archesporium forms hypodermally (Fig 4C). It differentiates directly into a megaspore mother cell (MMC) without of parietal cells formation. As result of meiosis running in MMC, a linear megaspore tetrad formed. The embryo sac (ES) development begins from the chalazal megaspore of the tetrad according to *Polygonum* (monosporic)-type. The mature ES consists of 3-celled egg apparatus with pyriform egg cell and two synergids (Fig 4D), a central cell and three antipodals. The synerdgids in mature ES are pyriform and hooked. The central cell forms before the fertilization, after the fusion of the two polar nuclei located near to the antipodals that are three, situated at the chalazal end of ES, resembling in shape and structure the egg apparatus but with smaller dimensions, and no proliferating.

The legitimate embryo and endosperm form after a porogamous double fertilization. The embryo formation follows the *Asterad*-type according Johansen classification [47]. The endosperm is cellular without passing through a free nuclear stage. In the seeds it is represented from one layer around the embryo (Fig 6E).

In *C. ruber* the formation of an embryo without suspensor situated deeply in the endosperm was observed (Fig 5B). Its topography suggests that it originates from the endosperm cells.

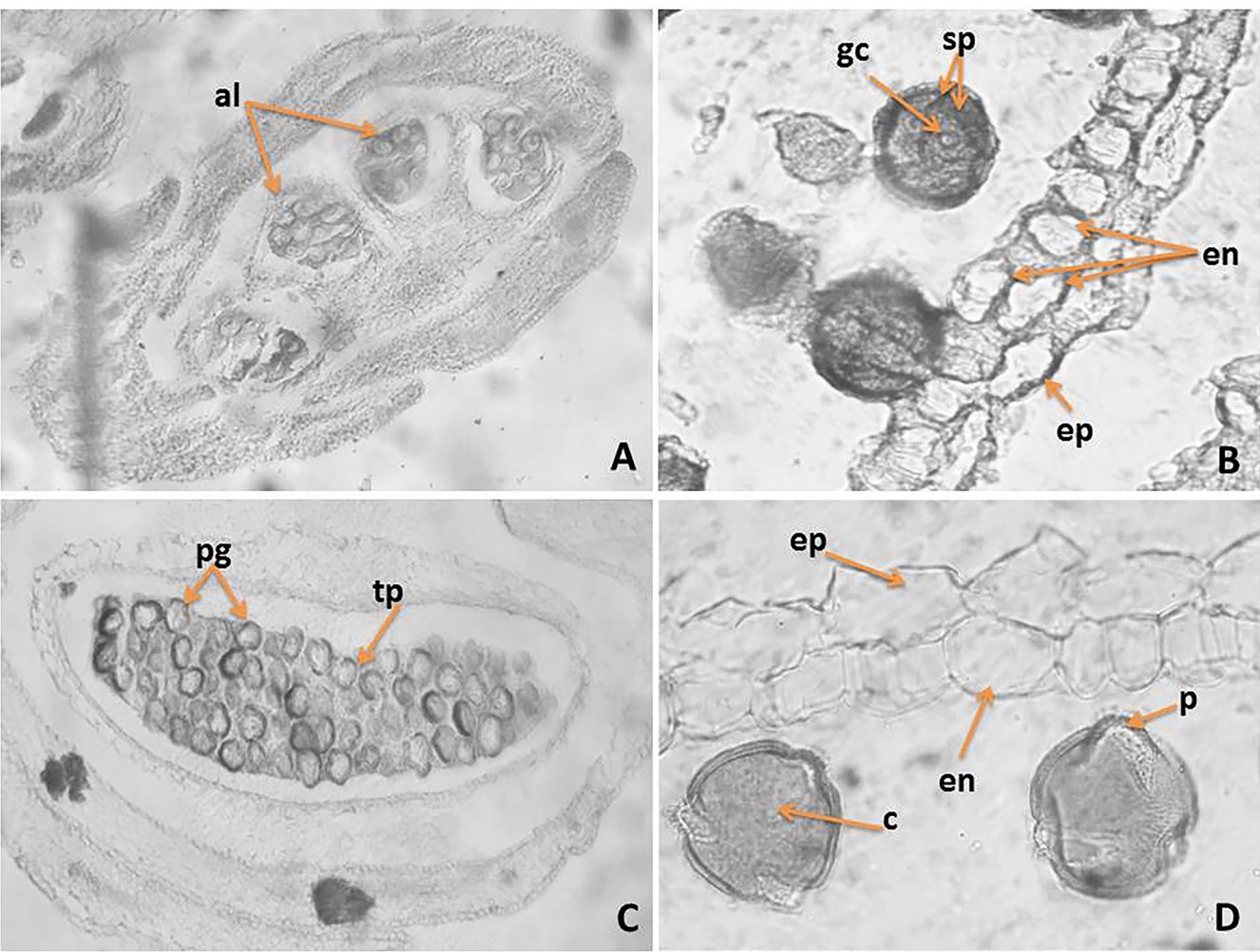

**Fig 3.** Anther and development of the male gametophyte: A– tetrasporangiate anther; B – two celled mature pollen grains and anther wall with two layered endotecium; C- pollen grains and ameboid tapetum; D – mature pollen grains and anther wall with epidermis and fibrous endothecium; gt–glandular trichomes, ep – epidermis, en – endothecium, al–anther locul, tp—tapetum, c—colpi, p—pori; pg—pollen gran. Magnification: A (x100); B,C,D (x400).

**3.3.3. Pollen viability.** As a result of the counting of the viable and nonviable pollen grains in the anthers of two studied species, the ratio of viable pollen was established to be 94.80 ±2.36% for *C. kellerei* and 92.09±2.45% for *C. ruber*.

**3.3.4. Seed viability.** On the base of the results of tetrazolium testing, the seeds from the two studied species were grouped in four classes (Figs 6 and 7):

Class I: seeds with embryos stained 100% in red (12% for the population of *C. kellereri* and 4% for the population of *C. ruber*, Fig 6A);

Class II: seeds with partially colored embryos (the root tip and part of cotyledons stained in red —30% for the population of *C. kellereri* and 10% for the population of *C. ruber*–Fig 6B and 6C);

Class III: seeds with colorless embryos (3% for the population of *C. kellereri* and 6% for the population of *C. ruber*–Fig 6D);

Class IV: empty seeds (52% for the two studied species-Fig 6E).

Following the criteria of Moore [37] for interpreting the resulting coloring from tetrazolium testing, the seeds from Classes I and II were estimated as viable, and the resulting percent

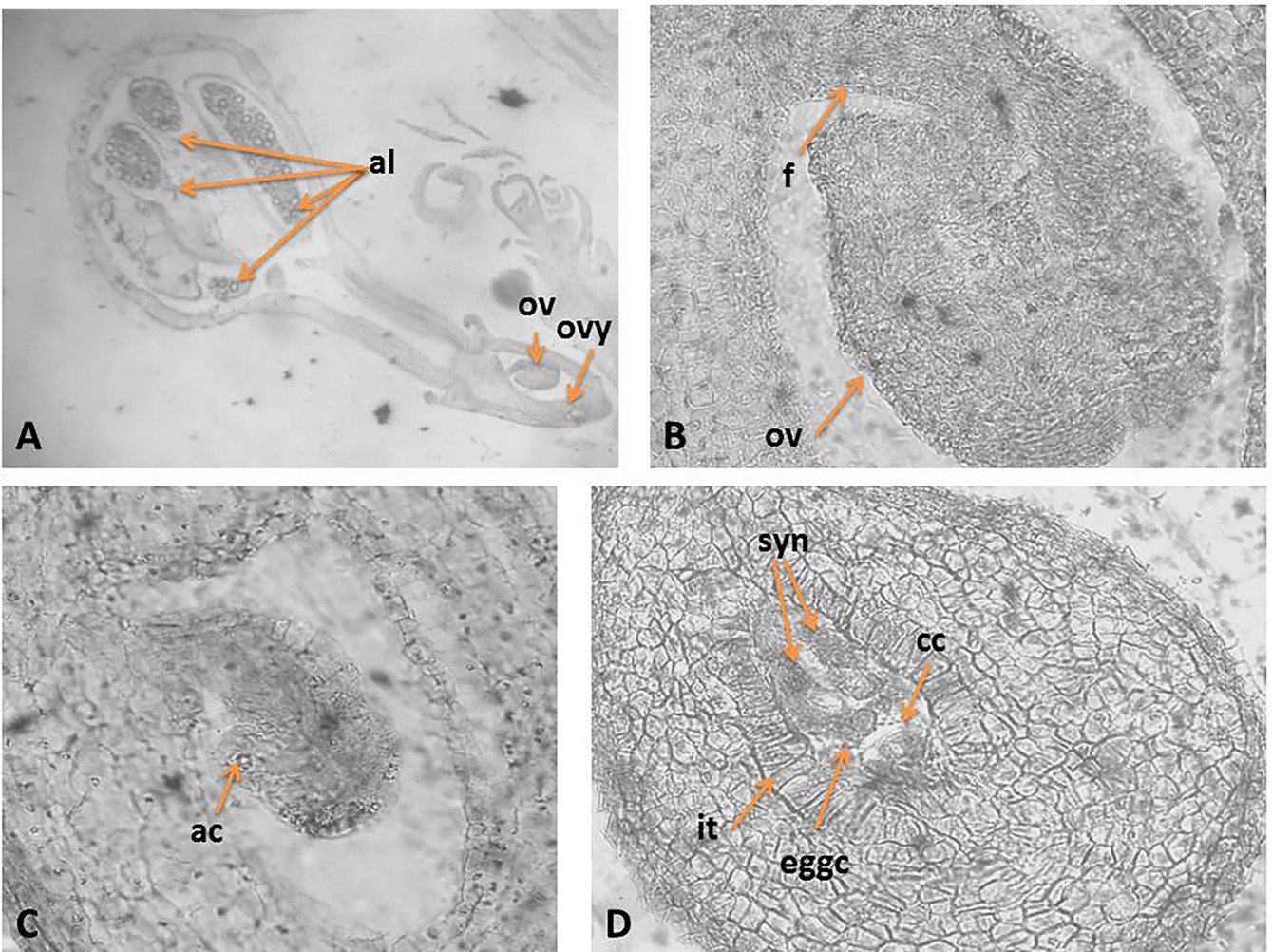

**Fig 4. Ovule and development of the female gametophyte.** A.–Flower with pistil and stamen; B -Anatropous unitegmic ovule; B. Mature *Polygonum*-type embryo sac (ES); C.–*Hemitropous* ovule with unicellular archesporium; D.—Mature ES with egg apparatus and central cell; ovy—ovary, ov – ovule, f—funiculus, it – integumental tapetum (endothelium), al—anther locul, ac—archesporial cell, eggc–egg cell, syn–synergids, cc–central cell. Magnification: A, (x100); B-D (x400).

of viable seeds were 42% for the population of *C. kellereri* and 14% for the population of *C. ruber* (Fig 7)

## 3.4. Micromorphological analysis by scanning electron microscopy (SEM)

The microstructure of the surface of the epidermis of the leaf, stem and the fruit of *C. kellereri* (population Bansko and Vratsa) and *C. ruber* were observed and analyzed by Scanning Electron Microscopy (SEM). Previously, it has been demonstrated that the micromorphology of the epidermis (shape and surface of the main epidermal cells; stomata; epicuticular waxes; striations) and fruit surface are important taxonomic characteristics [48–50]. These characteristics provide information for plant identification at submicroscopic level and are widely used by taxonomists to establish phylogenetic relationships [51, 52].

**3.4.1. Leaves surfaces of C. kellereri and C. ruber.**   The leaves of *C. kellereri* and *C. ruber* are simple, ovate-oblong to linear, oppositely arranged. The SEM analyses showed that the leaf surface is covered with a thick layer of epicuticular waxes. The waxes are arranged as threads and

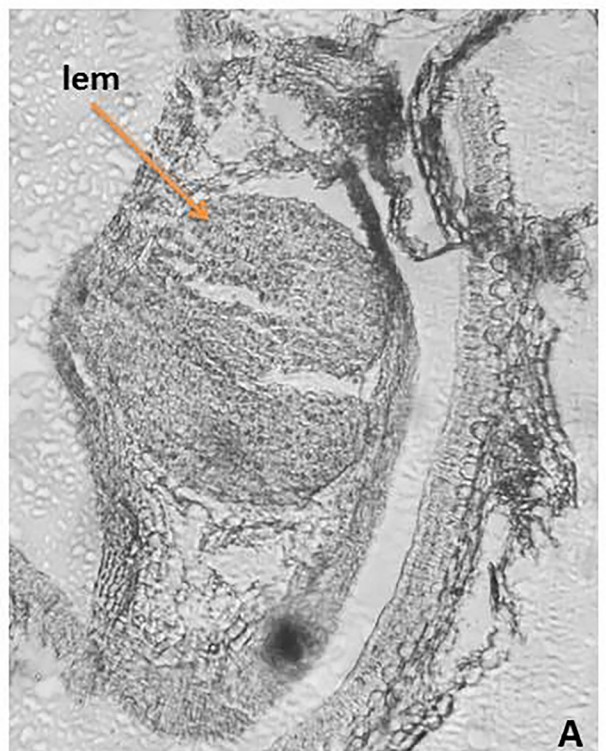
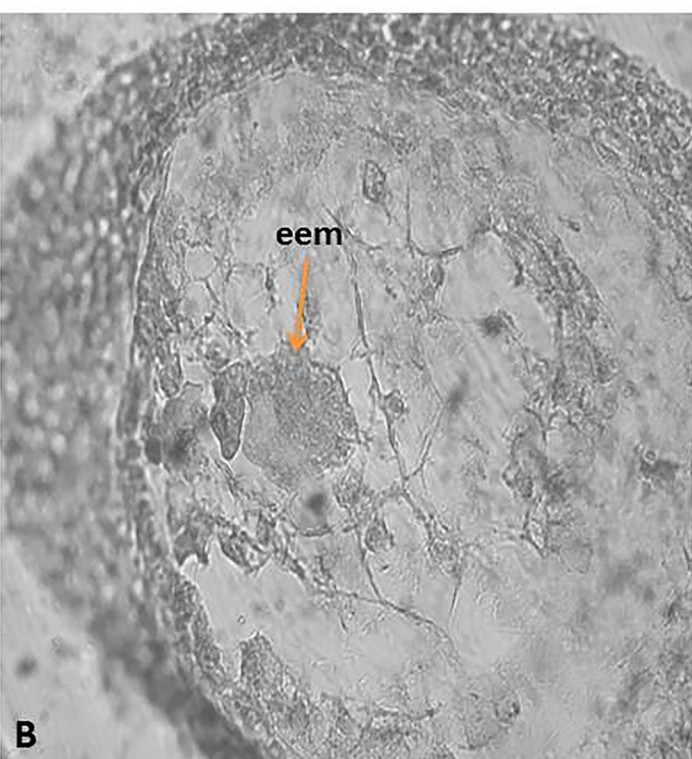

**Fig 5. Development of the female gametophyte of *Centranthus ruber*.** A.—Legitimate *Asterad*-type embryo; B—Endospermal embryo; lem—legitimate embryo, eem—endospermal embryo. Magnification: (x400).

form a network between the veins of the leaf, and on the veins themselves they are in a thick and continuous layer (Fig 8A and 8B). The main epidermal cells in *C. kellereri* are isodiametric-polygonal to elongated (Fig 8A and 8B), while the epidermal cells in *C. ruber* are isodiametric (Fig 9).

According to the terminology of Barthlott and Ehler [53] and Barthlott et al. [38] the surface of the periclinal walls of the leaves in both studied species is *Convex* type, with striations in *C. kellereri*. The anticlinal walls in *C. kellereri* are rounded-edged, striated, and vary from straight to S-shaped, convex [53]. The stomata are located on both epidermal surfaces (amphistomatic leaf). They are either immersed or not, depending on in which part of the epidermal surface they are located—in the convex part or in the concave part (Fig 8A and 8B).

At the edge of the stomatal closing cells of *C. kellereri*, the epicuticular waxes form minor stomatal chimneys (Fig 8B). A significant variation in the micromorphology of perioral cells was found in this study, both between the two *Centranthus* species and between the two populations of *C. kellereri* (Bansko and Vratsa) (Fig 8A and 8B). The cells from the stomatal complex in *C. ruber* are isodiametric with a smooth surface (Fig 9), while in *C. kellereri* these cells are isodiametric-polygonal to elongate with striations (Fig 8A and 8B).

In *C. kellereri* from Vratsa, the striations of the epidermal cells are oriented longitudinally or transversely on the cell surface (Fig 8A), while in the *C. kellereri* from Bansko, the striations are longitudinal to the cell (Fig 8B).

Depending on the arrangement of the striations on the cells from the stomatal complex of *C. kellereri* from Vratsa, the following varieties were established: stomatal complex where the cells were by transversely located striations (Fig 8A); stomatal complex where the cells were surrounded by longitudinal striations (Fig 8A); or stomatal complex where the cells do not

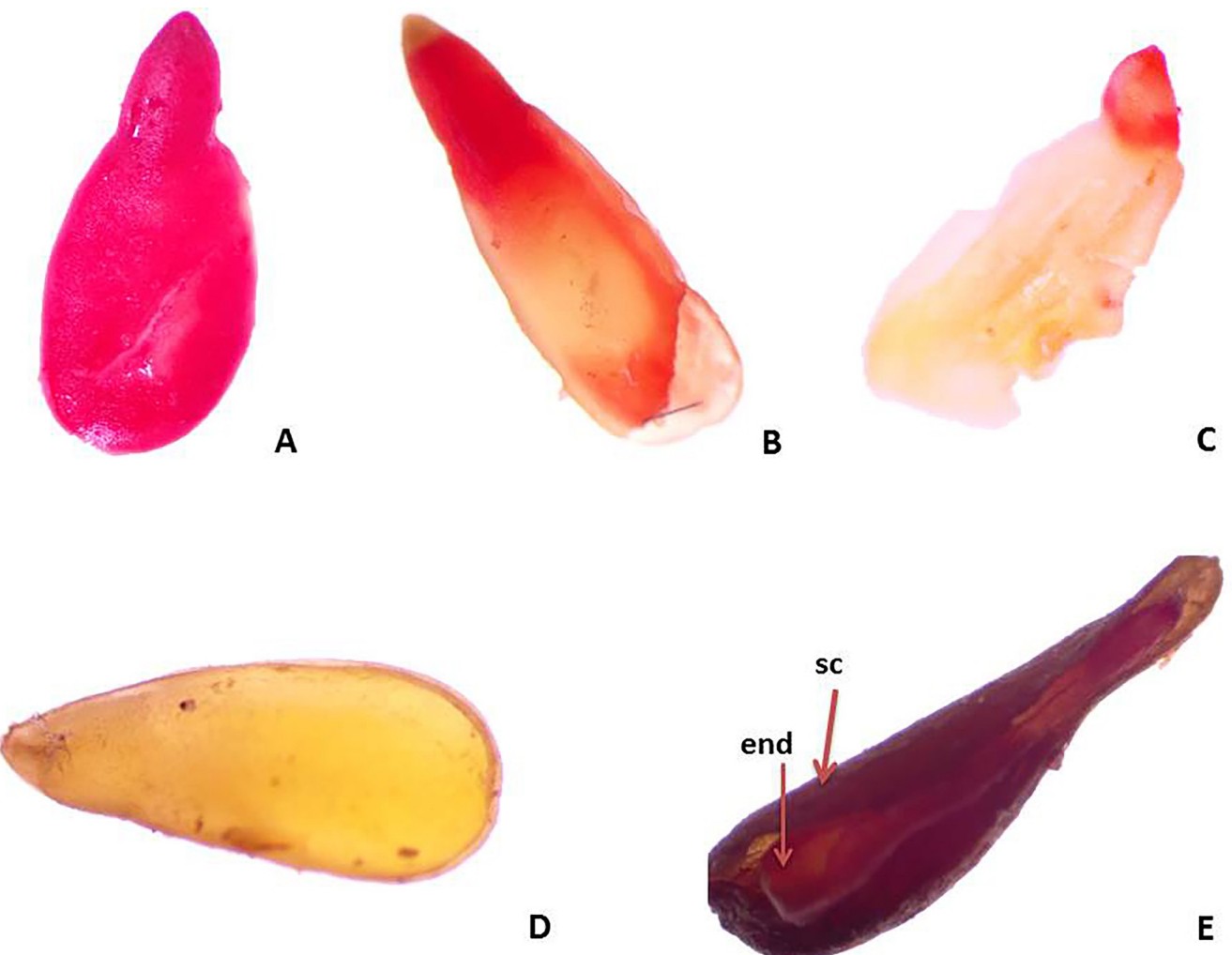

**Fig 6.** Estimation of seed viability according to Tetrazolium test: A – Viable, red colored embryo: B, C–Viable, partially colored embryos /B-colored root top and part of cotyledons; C–colored the root top/, stained in dark red /C/; D – Unstained unviable embryo; D—Unviable, empty seed (with seed coat and endosperm, and without embryo); end—endosperm, sc—seed coat.

have striations (Fig 8A). The cells in the stomatal complex of *C. kellereri* from the Bansko population are relatively constant and the striations are longitudinal. Single capitate glandular trichomes were found on the surface of *C. kellereri* (Fig 8A and 8B).

**3.4.2. Stem surfaces of C. kellereri and C. ruber.** According to Delipavlov et al. [14] the stem of *C. kellereri* and *C. ruber* is smooth and erect. From the SEM analyses performed in this study, it became clear that at micromorphological level, the epicuticular waxes form longitudinal ridges that give a ribbed shape to the stem (Fig 10). Stem waxes in *C. kellereri* differ both in shape and in arrangement. At the base of the stem, they form a continuous waxy crust layer, which continues along the convex part of the edges along the entire length of the stem.

Further up the stem in the areas between the edges of the waxy layer is thinner and crystalloids of various shapes, sizes and orientations are observed on it (Fig 10). According to the terminology of Barthlott et al. [38], platelets and plates crystalloids were observed in this study (Fig 10). The platelets were the most common crystalloids. The pattern of orientation of

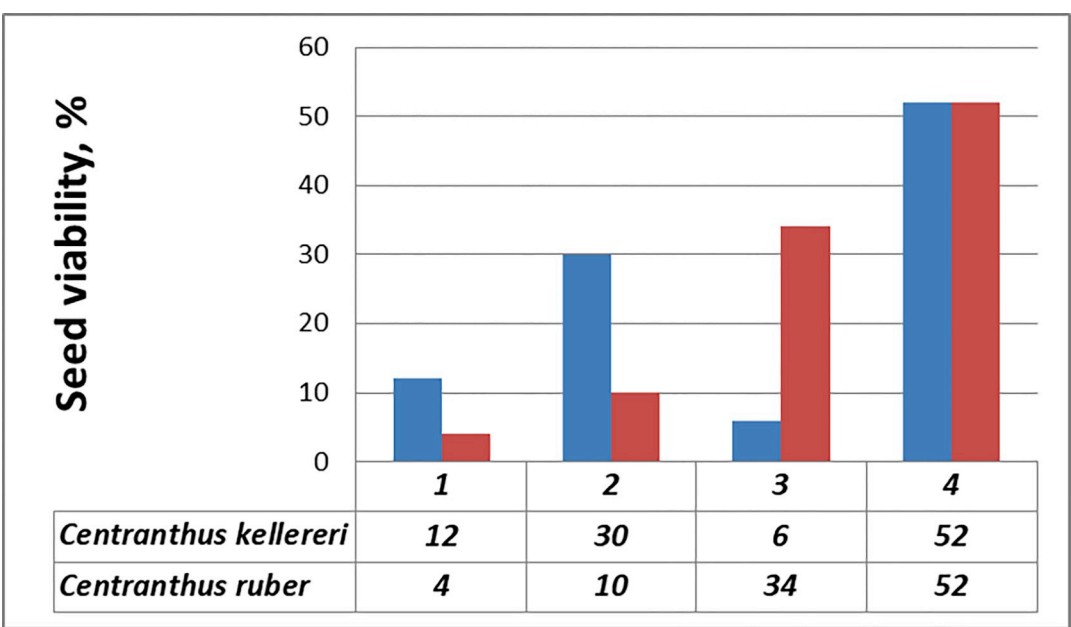

**Fig 7.** Frequency of seeds (embryos) viability (%) assessed by tetrazolium test: 1 –Class I, viable embryos (stained in red); 2 – Class II, viable embryos (partially colored embryos); 3 –Class III, unviable embryos (unstained); 4—Class IV, unviable (empty) seeds.

platelets was not uniform, as is the manner and angle at which they attach to the surface of the epidermis. For example, around the stomata they are locally isolated, while in other parts of the stem they are randomly oriented (Fig 10). The wax platelets also differ in the type of their surface and their edges, which are most often irregular and wavy. Plates were also established in separate segments between platelets. The arrangement of crystalloids, their orientation and density can be used as a taxonomic feature.

**3.4.3. Fruits surfaces of C. kellereri and C. ruber.** The micromorphological features of the fruit surfaces of *C. kellereri* and *C. ruber* analyzed by SEM are shown on Fig 11; Fig 12A and 12B. The fruits (an achene) in both species of *Centranthus* are egg-shape elongated with a dry pericarp which is fused with the seeds. In both species, the fruits have clearly formed dorsal and ventral sides. On the dorsal side of the fruit of both species of *Centranthus*, 5–6 longitudinal ridges are observed, and on the ventral side, there is only one ridge. In the front part of the fruit, where the sepals attach, a disc-like expansion (like a balloon) was observed. In the fruits of *C. ruber* this disc-like extension has characteristic semi-circular lobes (Fig 11), while in *C. kellereri* this extension has no definite shape but has numerous papillae (Fig 12A and 12B). Stomata were observed on the exocarp of *C. kellereri* fruits from both populations (Bansko, Vratsa) (Fig 12A and 12B).

The exocarp of the fruit of *C. ruber* is made up of cells with a rectangular rhombic shape and clearly differentiated anticlinal walls (Fig 11). If we use the terminology of Barthlott and Ehler [53], the surface of the periclinal walls in the fruit of *C. ruber* is *Tabular* type to slightly *Convex* type (Fig 11).

The cells of the exocarp in *C. kellereri* from both the Bansko and Vratsa populations are diamond-shaped, and in some parts of the fruit have unclear anticlinal walls (Fig 12A and 12B). Around the edges of the fruit, the anticlinal walls in this species are significantly elevated compared to the rest of the fruit surface. If the terminology defined by Barthlott and Ehler [53] is used, then the surface of the periclinal walls in the fruit of *C. kellereri* is *Tabular* type and only

**Fig 8. A.** Scanning Electron Microscopy (SEM) analysis of leaves surfaces of *C. kellereri* from Vratsa (V); ad–adaxial surfaces; ab–abaxial surfaces; view of trichomes, epicuticular waxes, stomata and striations. **B.** Scanning Electron Microscopy (SEM) analysis of leaves surfaces of *C. kellereri* from Bansko (P); ad–adaxial surfaces; ab–abaxial surfaces; view of trichomes, epicuticular waxes, stomata, striations and stomatal chimneys.

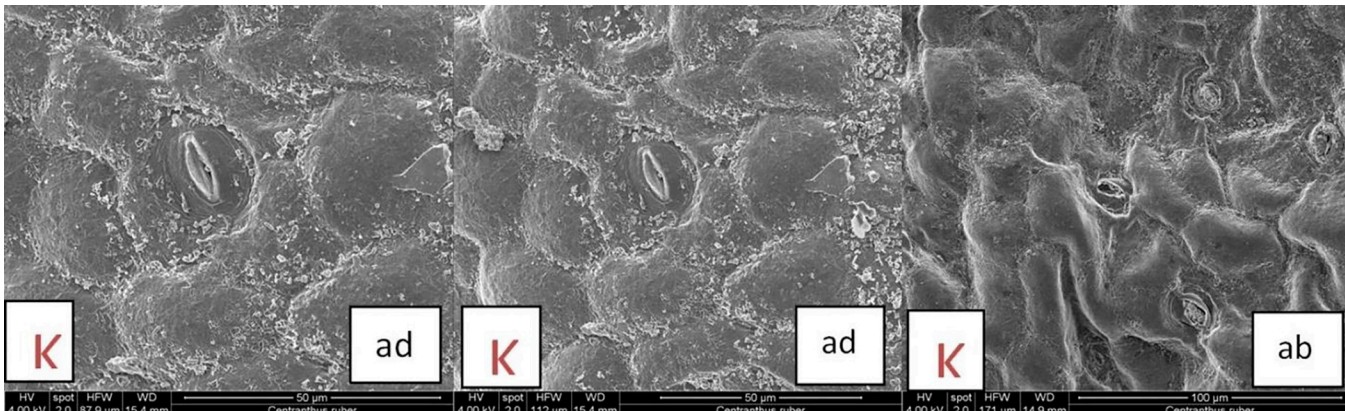

**Fig 9.** Scanning electron microscopy (SEM) analysis of leaves surfaces of of *C. ruber* from Kazanlak (K); ad–adaxial surfaces; ab–abaxial surfaces; view of epicuticular waxes and stomata.

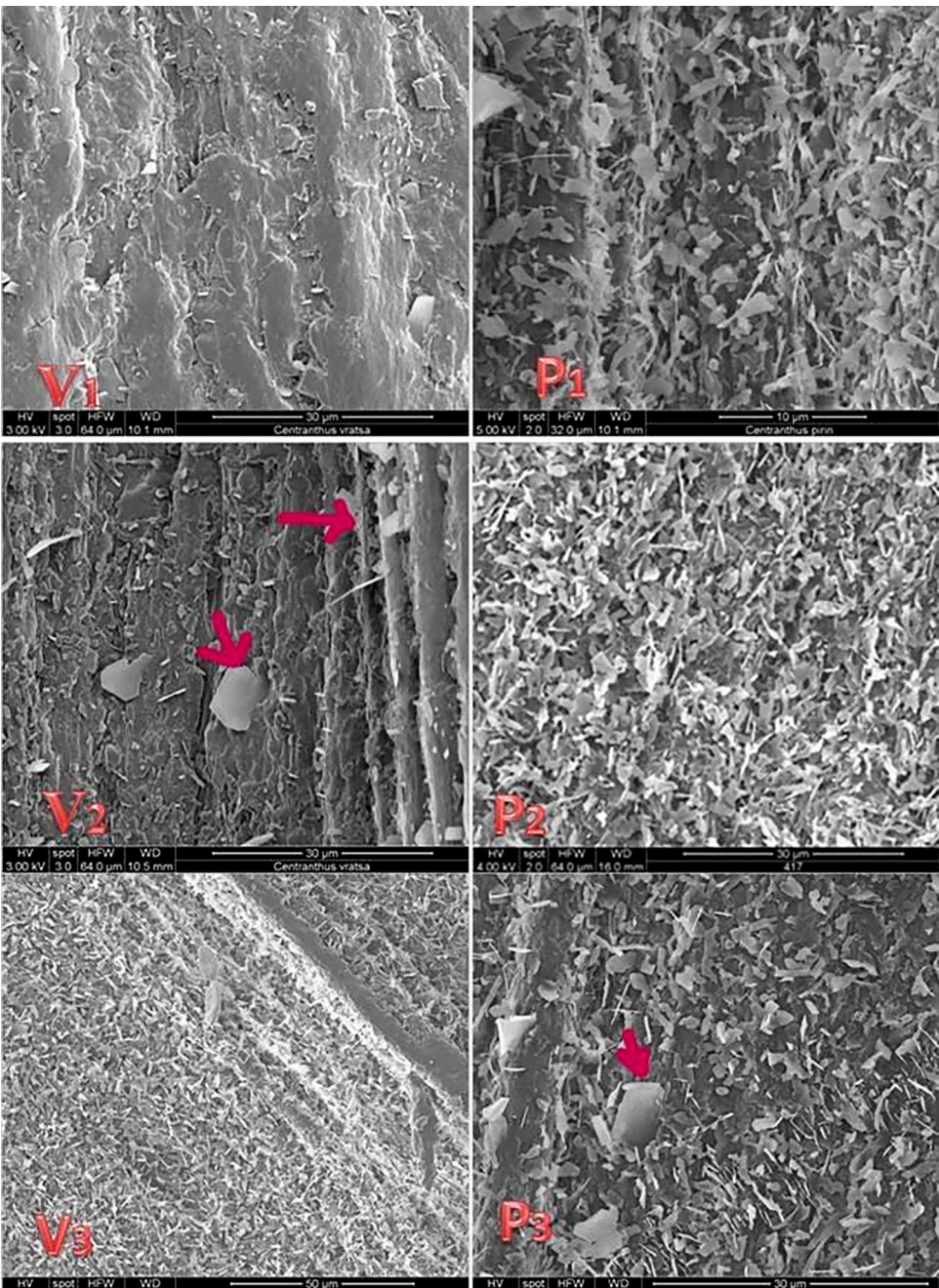

**Fig 10. Scanning electron microscopy (SEM) analysis of stem surfaces of *C. kellereri* from Bansko (P) and Vratsa (V). View of waxes, platelets and plates.**

around the edges it becomes slightly *Convex* type (Fig 12A and 12B). Papillae and short hairs were observed in fruits from both populations where the carpels fuse and in the expanded part of the margins (Fig 12A and 12B).

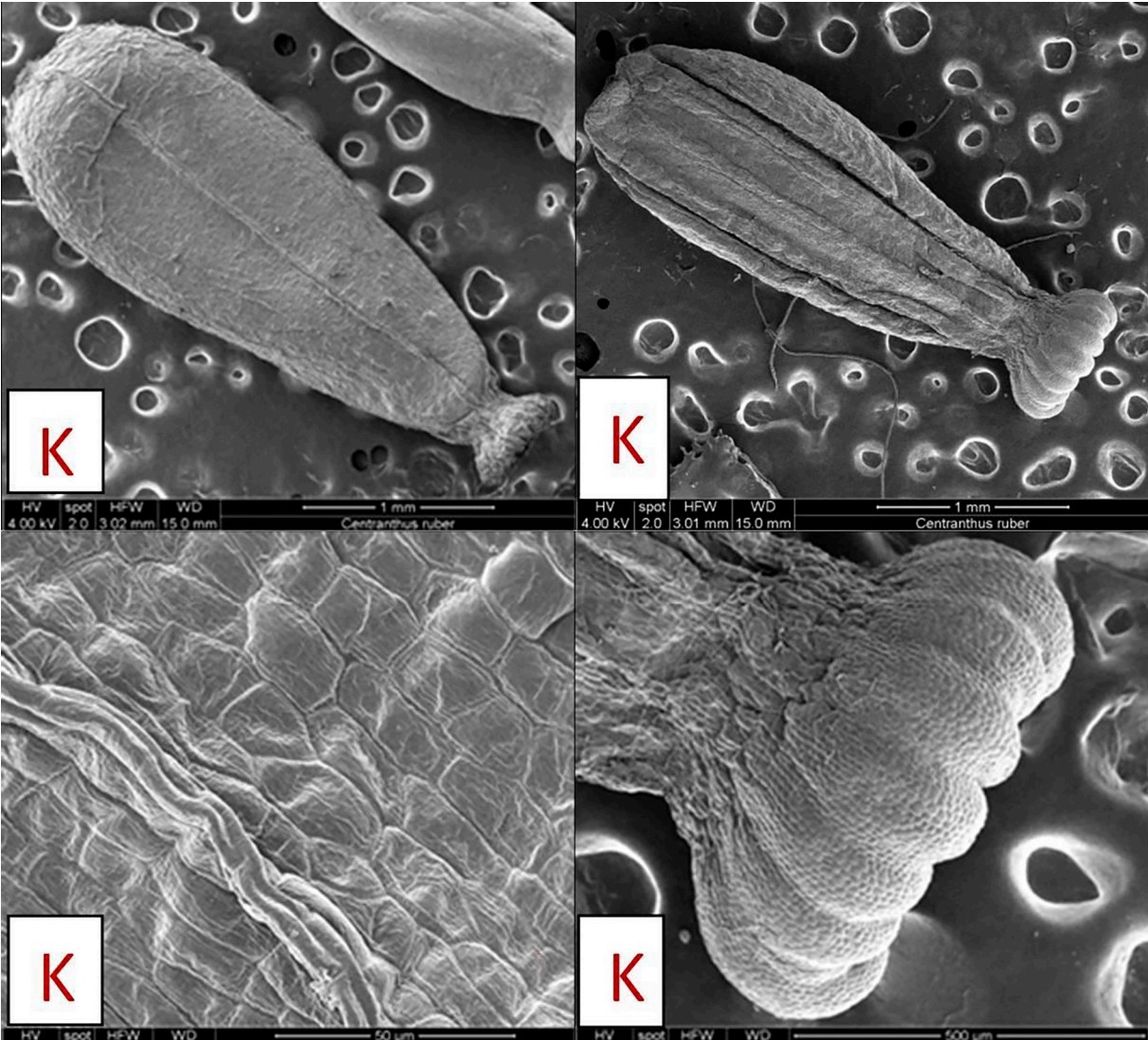

**Fig 11. Scanning electron microscopy (SEM) analysis of fruits surfaces of *C. ruber* from Kazanlak (K). General view of exocarp of fruits.**

## 3.5. Phytochemical analyses

Metabolomic analyses of the extracts from aboveground and root samples of the *Centranthus* species revealed a number of compounds (Fig 13). Screening of extracts of the roots and aboveground plant parts against the Enzo Life Sciences Natural Products Library [54] revealed the presence of 32 compounds that were tentatively identified as 6 simple phenolics (caffeic acid, scopoletin, chlorogenic acid, vanillyl acetone, rosmarinic acid, shikimic acid), 18 flavonoids (flavokawain A, eriodictyol-7-*O*-glucoside, marein, dihydrorobinetin, kaempferol-7-*O*-neohesperidoside, apigenin, apigenin-7-*O*-glucoside, luteolin, maritimein, isoquercitrin, eriocitrin, diosmin, narirutin, isorhoifolin, naringenin, rutin, isorhamnetin-3-*O*-glucoside, isorhamnetin-*O*-rutinoside), 1 quinone (emodin), 1 lipid (phytosphingosine), 1 alkaloid (stachydrin), 2 diterpenes (lagochiline, sclareol), and 3 triterpenoids (ursolic acid, betulinic acid, oleanolic acid). Fig 13 shows a heatmap with the distribution of the phytochemicals

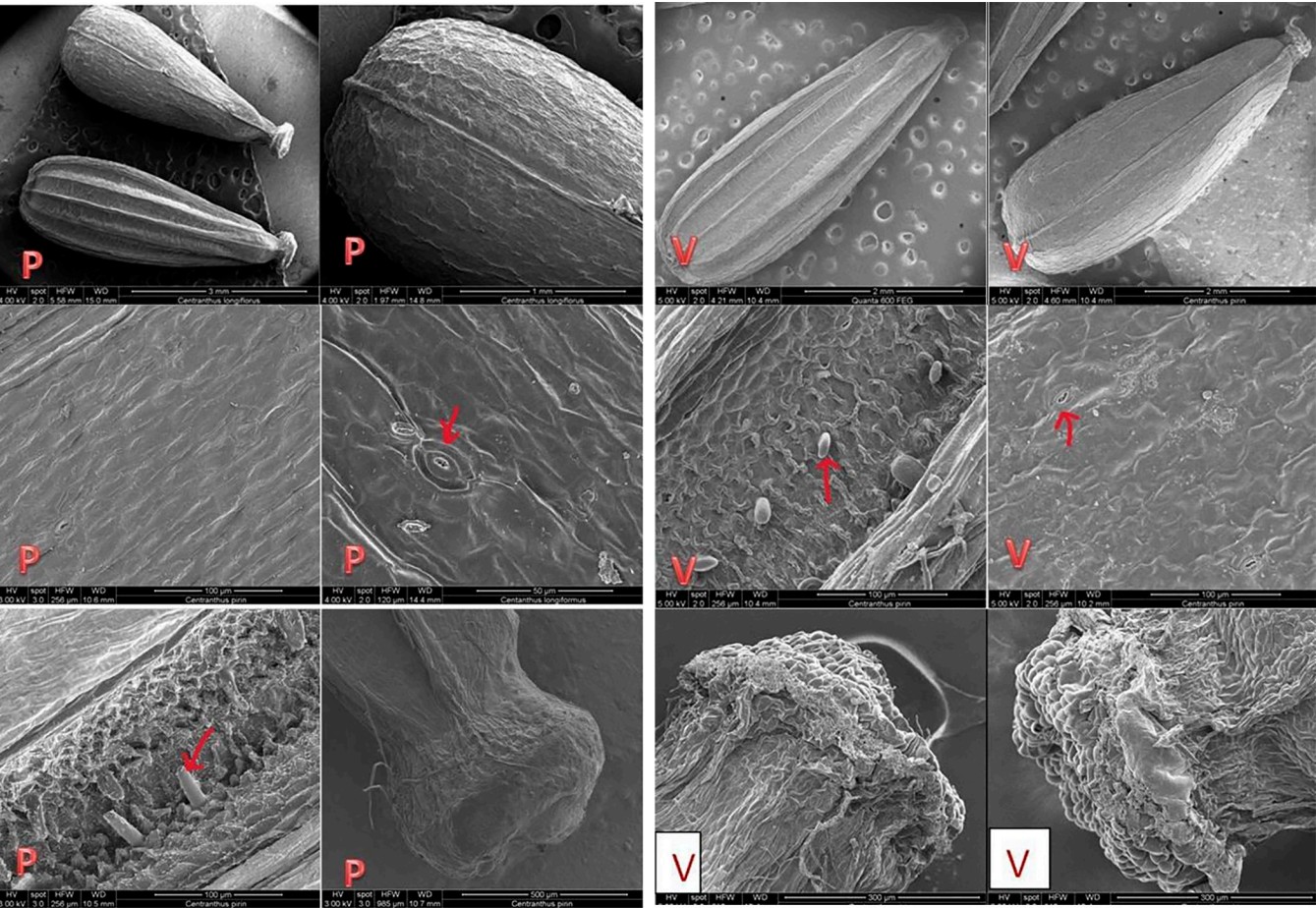

**Fig 12. A.** Scanning Electron Microscopy (SEM) analysis of fruits surfaces of *C. kellereri* from Bansko (P). General view of exocarp of fruits and trichomes, papili, stomata. **B.** Scanning Electron Microscopy (SEM) analysis of fruits surfaces of *C. kellereri* from Vratsa (V). General view of exocarp of fruits and trichomes, papili, stomata.

across the species, plant parts, and location. Roots and shoots differ phytochemically much more than the two species or accessions from different harvest locations (Fig 13).

### 3.6. Antioxidant activity

*Total phenols and flavonoids*. The means of the studied indicators are presented in Table 12. Overall, of the three extractants, 70% ethanol showed the strongest extraction properties for total phenolics and flavonoids, followed by water and finally acidic methanol.

The 70% ethanol extract of *C. ruber* had the highest polyphenol concentration (7.976 mg GAE/g DW), followed by *C. kellereri* from Bansko and Vratsa (with 6.382 and 6.331 mg GAE/g DW, respectively).

With the same extractant, the highest values for flavonoids were reported at *C. kelleri* from Vratsa (302.9 mg QE/g DW), followed by *C. ruber* and *C. kelleri* from Bansko (226.2 and 220.7 mg QE/g DW, respectively).

*Free radical scavenging activity*. The antiradical activity of the extracts was measured with ABTS and DPPH against the Trolox standard. The results demonstrated that aqueous and ethanol extracts in all populations show approximately the same ABTS radical scavenging activity.

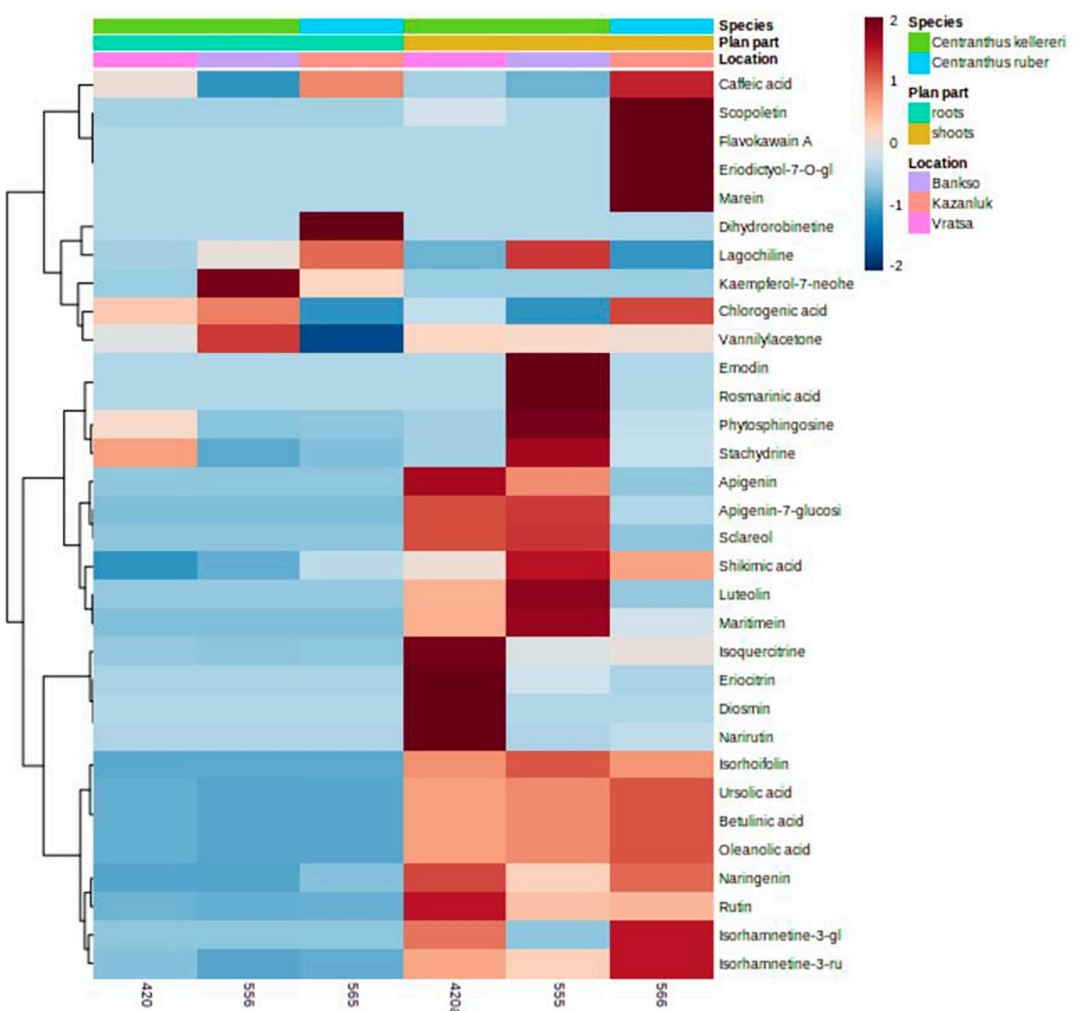

**Fig 13. Heatmap showing distribution and relative abundance of phytochemicals in *Centranthus* spp.** The red and blue colors represent abundance for each phytochemical relative to its mean set at zero. Some of the compound names are truncated due to character limitations; full names of compounds can be found in the Results section.

Only acid methanol extracts had statistically lower activity, being lowest in the Kazanlak population (0.667 mg TE/g DW) (Table 12).

With regard to DPPH radical scavenging activity, in the *C. kellereri* population of Vratsa, all three extracts gave approximately the same values; however, the values were not statistically different (0.264, 0.253, and 0.235), but they turn out to be higher than the results obtained in the other populations. The extracts obtained from the *C. ruber* Kazanlak population showed the weakest radical scavenging activity.

## 4. Discussion

### 4.1. Genetic diversity and differentiation

The ISSR data showed moderate levels of genetic diversity across the populations of *Centranthus* species in Bulgaria, as reflected in the values of, for example, genetic diversity and Shannon's information indices (Table 13). The prevalence of sexual reproduction and relatively good reproductive capacity explain the genetic diversity found in *Centranthus*

populations. On the other hand, the restricted distribution may also affect the genetic diversity. Therefore, the observed moderate levels of genetic diversity in *Centranthus* populations could be explained by the putative effect of reduced gene flow among populations, since the habitats are separated by inter-mountains valleys and mountains. In addition, private alleles were noted (Table 13). Most probably, the existence of private alleles was due to the restricted gene flow. In our opinion, the topology of localities, as well as the geographical distance between investigated population acts as a barrier to seed and pollen dispersal. The spatial discontinuities are common in *Centranthus* populations because of their specific habitat requirements and they act simultaneously as a temporal barrier to pollen-mediated gene flow arising from a significant difference with spatial isolation between the habitats and enhance local adaptation. Our assumptions were also supported by the PCoA (Fig 1) and structure (Fig 2). This geographical isolation in the *Centranthus* populations typically restricts gene flow leading to reduction of genetic diversity and increasing the differences between investigated populations. In the longer term, if the populations of the species remain small, there is a potential for further decrease in their level of genetic diversity that may cause a reduced fitness and lack of species' adaptability to the changing environments.

The maintenance of moderate diversity in *Centranthus* populations, even the presence of a significant geographic distance between populations today, probably is a result of the maintenance of high genetic diversity at a historical moment of population connectivity.

The estimate of gene flow obtained in this study (Nm = 0.899) is not high and cannot prevent further differentiation through genetic drift, although, the result of AMOVA revealed that *Centranthus* populations showed some good level of within-population genetic variability (66% of their total genetic variation). This should be related to the out-crossing nature of *Centranthus* species. Our findings suggest that the present-day distribution of *C. kellereri* and *C. ruber* has likely been limited to ecological factors and habitat specialization.

Understanding genetic diversity and population structure of *C. kellereri* and *C. ruber* in Bulgaria is critical for effective management of its genetic resources. Genetic variation has increasingly been recognized as crucial to success in the long-term management of species and therefore should be a central concern for the long-term conservation. From the results revealed in our study, the levels of genetic diversity of *Centranthus* populations have reduced, we assess that the main threats are the conditions of its habitats, population specialization, as well as the anthropogenic impact.

## 4.2. Embryological research

The present study is the first one on the peculiarities of the reproductive biology of Bulgarian population of *C. ruber* and *C. kellereri* that contributes for the embryological characteristic of genus *Centranthus*. Most of the revealed characteristics of the structures in male and female gametophytes are in accordance with that described for the representatives of Valerianaceae family in the actual Plant List [55], which is placed as subfamily in the Caprifoliaceae family, namely: tetrasporangiate anthers, 4-layered anther walls, ameboid tapetum, normal meiosis of the simultaneous type, anatropous tenuinucellate ovule, *Polygonum*-type development of female gametphyte with hooked synergids and three noproliferating antipodal cells with arrangement similar to the cells of egg apparatus, *Asterad*-type of embryo formation [56–58].

The established *C. ruber* endospermal embryo is evidence of higher plasticity of the female gametophyte, which provides greater adaptability to environmental changes and, correspondingly, stability of the populations of the species. It is a new feature for genus *Centranthus*. For this genus only formation of endospermal haustorium in *C. calcitrapa* was reported [58] and for Valerianaceae family a poliembriony [58].

Mattana et al. [59], by comparing the germination ecology of *C. ruber* and the endemic for Sardinia *C. amazonum*, established lower seed germination percentage and more empty seeds in *C. amazonum* than in *C. ruber*. The results of the present study revealed the contrary: a higher percentage of seed viability was established in the seeds of *C. kellereri*. This fact corroborates what was also reported in species from other families: dependence of seed quality and germination on the environmental conditions. For example, Estrelles et al. [60] established that the seed germination behavior in two endemic species of the genus *Sideritis* was closely related to the characteristics of habitats, mainly the environmental conditions (temperature and sunlight intensity) and soil structure. Generally, the established low (12% in *C. ruber*) to moderate (42% in *C. kellereri*) seed viability is one of causes for limited distribution of two studied species in Bulgaria.

## 4.3. Morphological analysis by scanning electron microscopy (SEM)

*C. kellereri* is an endemic plant species found only in Bulgaria [61], classified as Critically Endangered ([CR B2ab(v)]) [5]. However, the micromorphological features of its leaves, stems, and fruits have not been previously studied. Therefore, this is the first study to describe the characteristics of the epicuticular waxes present on the leaves and stems of this species. Furthermore, the study also revealed the characteristics of the exocarp in the fruits of *C. kellereri* and compared them with those in *C. ruber*. The morphological characteristics of the flowers, fruits, microscopic structure of the epidermis (on the leaf, stem, and fruits), and pollen in species belonging to the genus *Centranthus*, are of taxonomic significance [62–64].

These traits are characterized by a high degree of variability [63–66], which is a prerequisite for the accumulation of taxonomic synonymy and interpretations. The SEM analysis of the leaf epidermis of the two *Centranthus* species showed similarities but also significant differences in micromorphological characteristics. Differences in the orientation and striations of the cells of the stomatal complex in *C. kellereri* were set between samples from the two populations (Bansko and Vratsa). The micromorphology of the stomatal complex cells of *C. kellereri* from Vratsa was characterized by a greater diversity, which defines four types of the stomatal complex cells. Stem epicuticular waxes in *C. kellereri* also showed diversity, in the two populations, (Bansko and Vratsa), both in shape and arrangement. *C. kellereri* is known as a calciphyte and spreads on mobile stony screes (Calcareous screen) characterized by a low amount of surface water (sclerophyte). Most probably, the formation of a thick layer of the crust of epicuticular waxes and crystalloids is a reaction of the species to unfavorable conditions. Normally, plants react by synthesizing large amounts of epicuticular waxes to protect themselves from high light intensity and insufficient water.

The fruits of both species of *Centranthus* are ovoid elongated with a pappus-like disc-like extension and are morphologically similar to the fruit of *Valeriana* sp. Indeed, the genus *Centranthus* was included in tribe Valerianeae, subtribe *Centranthinae* Graebn. in Valerianaceae family. Although according to the recent taxonomic developments [15–17] the family is referred to the family Caprifoliaceae Juss. (*sensu lato*). Papillae, short hairs and stomata were found on the surface of *C. kellereri*, which have also been found in other *Centranthus* species [67, 68].

## 4.4. Essential oils composition of *C. kellereri* and *C. ruber*

Although no studies on the composition of *C. kellereri* EO could be found, some have been conducted on *C. ruber*. In a study by Musolino et al. [11], aboveground plant parts of *C. ruber* from Southern Italy were extracted with methanol via maceration, and the crude extracts were subsequently separated using solvents with increasing polarity. The dried extracts were

suspended in a solution of 9:1 methanol and water, and then extracted with n-hexane. The dried residue was extracted with dichloromethane. The authors reported the presence of chemical compounds in the n-hexane and dichloromethane fractions. Fatty acids (FA) were found to be abundant in the n-hexane fraction, with palmitic acid being the most abundant (6.87%). The GC analyses also identified four terpenes, including the monoterpene lactone dihydroactinidiolide and the isoprenoid ketone phytone (hexahydrofarnesyl acetone), and the monoterpenoid 2,6-di-tert-butyl-1,4-benzoquinone, with phytone being the most abundant at 2.8%. Six compounds were identified in the dichloromethane fraction of *C. ruber*, with benzoic acid being the most abundant (1.2%), and the rest present in trace amounts, mostly below 0.5% [11].

## 4.5. Phytochemical analyses

The KnapSack database [69] of plant species and their phytochemicals reveals that only three species of *Centranhus* have been characterized phytochemically: *C. macrosiphon*, *C. longiflorus*, and *C. ruber*. Our plant metabolomics approach did not detect the previously reported acevaltrate [70] in the one accession of *C. ruber* because this phytochemical is not present in our Enzo library of plant metabolites. The study focused on valepotriates and the author did not report the phytochemicals we detected in this study. Consistent with the finding reported by Pagani [71] for *C. ruber*, we detected quercetin, rutin, kaempferol, luteolin, and their *O*-glycosides in the shoots of this plant.

## 4.6. Antioxidant activity of *C. kellereri* and *C. ruber*

In a previous study, *C. longiflorus* was shown to possess sedative and anticonvulsant effects, which were analogous to those exhibited by diazepam [22]. In previously taxonomical schemes, the genus *Centranthus* was considered to belong to the Valerianaceae family [14]. Usually, these species are utilized for their sedative properties [72–74]. However, the information on the chemical composition of the roots is scarce and is mainly related to the iridoids characteristic of this genus and in particular valepotriate. There is almost no information concerning antioxidant activity and chemical content of *C. kellereri*, which is Bulgarian endemic species.

This study compared two species, *C. ruber*, which is widespread, and the Bulgarian endemic species *C. kellereri*, from the only two existing populations in the world. The healing properties of wild plants are mainly due to their accumulated secondary metabolites, which, in turn, depend on the environmental conditions in which the plants grow. As expected, the results obtained from the studied populations differed significantly.

The method of extraction of biologically active substances is also of great importance. The use of different solvents, extraction times, or environmental conditions can significantly affect the results. In the present study, it was observed that ethanol and aqueous extracts had the highest values for phenolic and flavonoid compounds. Most of the extracts used for medical purposes are obtained with ethanol or water [72, 75]. Regarding aqueous extracts, similar results have been reported by other authors who compared the effect of solvents on the content of phenols and flavonoids in *C. longiflorus* [76].

Polyphenols are prominent due to their antioxidant activity and protective properties. The ethanolic extract exhibited the highest total phenolic content (7.976 mg GAE/g DW) in *C. ruber*. Conversely, methanolic extract (0.585 mg GAE/g DW) showed the lowest phenolic content in *C. ruber*. The last finding contradicts the widely spread opinion for methanol as an effective solvent for the extraction of antioxidants [77].

Flavonoids show several antioxidants, antiviral, and antimutagenic effects. For example, quercetin is a well-known plant-derived flavonoid that may have anti-inflammatory and anti-oxidant properties. Our results on flavonoid content were highest in *C. kellereri*, Vratsa, com-pared to the other populations, and that was observed for all extracting solutions in the following order: ethanol>water>acidic methanol (302.9; 208.6; 156.2 mg QE/g DW). In addi-tion, the results were higher than that previously reported in *C. longiflorus* [76].

DPPH radical (2,2-Diphenyl-1-picrylhydrazyl) is one of the few stable organic nitrogen rad-icals, which has a deep purple color. Unlike ABTS, it does not have to be generated before the assay. The DPPH assay is mainly based on an electron transfer reaction and hydrogen-atom abstraction. This assay is based on the measurement of the reducing ability of antioxidants toward DPPH [76]. *C. kellereri* Vratsa population showed the highest radical scavenging activ-ity in all extracts, which did not differ significantly, and were close to those obtained from the *C. kellereri* Bansko population. The population of *C. ruber* showed significantly lower scaveng-ing activity. Interestingly, the reported results for methanolic extracts did not correspond to those obtained from other authors [76].

The scavenging ability of the ABTS radical was found to be higher than that of the DPPH radical. Surprisingly, the water extracts obtained the highest ABTS radical scavenging activity in the order regarding the populations: Vratsa>Kazanlak>Bansko. This is the first study on the antioxidant activity of different extracts from *C. kellereri*. Different extracts (ethanolic, acidic methanol, and water) were analyzed in order to obtain the total phenolics, flavonoids, and radical scavenging activities.

Given the global popularity of the closely related species *C. ruber* as a landscaping plant [78, 79], and the exceptional ability of the endemic species *C. kellereri*, to thrive in inhospitable rocky screeds, there is significant potential for the endemic species *C. kellereri* to serve not only as a medicinal resource but also as a highly valuable addition to landscaping projects. Its inclusion in landscaping designs can contribute to the creation of visually appealing and sus-tainable landscapes, capable of withstanding challenging conditions. Furthermore, the utiliza-tion of *C. kellereri* in landscaping projects not only adds aesthetic value but also offers an opportunity to conserve and showcase the endemic flora, highlighting its ecological significance.

Considering these factors, it is evident that the endemic species *C. kellereri* holds tremen-dous promise, both in terms of medicinal applications and as a valuable asset in landscaping endeavors. Exploring and promoting its use in these domains can foster biodiversity conserva-tion and present novel possibilities for sustainable landscaping practices.

## 5. Conclusions

This is the first comprehensive study on *C. kellereri*, an endemic plant found only in two loca-tions in the world. Genetic analyses have shown a clear differentiation between *C. kellereri* and *C. ruber*, and between the two populations of *C. kellereri*. Phytochemical analyses have revealed both similarities and noteworthy differences in the content and composition of essential oils (EO) between *C. kellereri* and *C. ruber*, as well as within the two populations of *C. kellereri*. These differences include significant variations in the composition and concentration of indi-vidual compounds, which are influenced by factors such as plant part and timing of sample collection.

Further metabolomic analyses identified the presence of 32 compounds, many of them shared by the shoots and roots of both *C. kellereri* and *C. ruber*. Furthermore, variations in the detected compounds were observed between the *C. kellereri* plants at different locations, as well as between the roots and shoots.

This study represents the first investigation into the antioxidant activity of various extracts derived from *C. kellereri*. Our analysis included ethanolic, acidic methanol, and water extracts, aiming to determine the total phenolic and flavonoid content, as well as the radical scavenging activities. The *C. kellereri* Vratsa population exhibited the highest radical scavenging activity across all extracts. Notably, the scavenging activities did not exhibit significant differences and were comparable to those obtained from the *C. kellereri* Bansko population. In contrast, the *C. ruber* population displayed significantly lower radical scavenging activity when compared to both *C. kellereri* populations. These findings shed light on the antioxidant potential of *C. kellereri* and suggest that the Vratsa population, in particular, holds promise for further exploration. The results also highlight the need for additional research to understand the underlying factors contributing to the observed differences in antioxidant activity among these populations.

Regarding the established peculiarities of the structures and processes in the male and female gametophytes of the two studied *Centrantus* species, it may be defined by the following conclusions: The observed characteristics define the *C. kellereri* as sexual reproducing. Specifically, (1) The estimated normal running of processes leading to pollen and seed formation without deviation and degenerations and high pollen viability afford *C. kellereri* a high potential for realization of its reproductive capacity that is the prerequisite for regeneration and stability of their populations, and (2) The observed low seed viability indicates the *C. kellereri*'s high sensitivity to environmental conditions, which serves as a crucial factor affecting population size. Therefore, it is recommended to monitor the populations of *C. kellereri*, establish a seed bank, and implement experimental ex situ collections to propagate and reinforce their natural populations as needed.

The SEM analyses revealed notable variations in surface microstructural traits not only between the species (*C. kellereri* and *C. ruber*) but also within the two populations of *C. kellereri*. Consequently, the observed dissimilarities in genetic makeup, micromorphological characteristics, and phytochemical composition strongly indicate that the two populations can be classified as distinct subspecies or forms (or varieties) of *C. kellereri*; var. *pirinensis* and var. *balkanensis*. These findings underscore the importance of recognizing and differentiating between these populations, as they likely possess unique biological and ecological attributes. Further research is warranted to elucidate the specific characteristics and potential implications of these subspecies/forms/varieties in the context of their taxonomy, conservation, and utilization.

Considering the widespread use of the closely related species *C. ruber* as a landscaping plant worldwide and the fact that the endemic species *C. kellereri* thrives in rocky screes under extremely challenging environmental conditions, there is a strong potential for the endemic species *C. kellereri* to be utilized not only for medicinal purposes but also as a valuable landscaping plant.

## Supporting information

**S1 File.**
(DOCX)

## Acknowledgments

The authors extend their gratitude to the Directorate of Pirin Park and the Bulgarian Ministry of Environment and Water (MEW) for granting special permit (736/12.032018) issued to Dr. V.D. Jeliazkov (Zheljazkov), Mrs. Daniela Borisova, and Dr. C. Radoukova. This permit enabled the collection of plant tissue samples vital for this study.

## Author Contributions

**Conceptualization:** Valtcho D. Zheljazkov.

**Data curation:** Valtcho D. Zheljazkov.

**Formal analysis:** Valtcho D. Zheljazkov, Ivanka B. Semerdjieva, Elina Yankova-Tsvetkova, Lyubka H. Koleva-Valkova, Galya Petrova, Ivayla Dincheva, Fred Stevens, Wenbin Wu, Tess Astatkie.

**Funding acquisition:** Valtcho D. Zheljazkov.

**Investigation:** Valtcho D. Zheljazkov, Ivanka B. Semerdjieva, Daniela Borisova, Elina Yankova-Tsvetkova, Lyubka H. Koleva-Valkova, Galya Petrova, Ivayla Dincheva, Fred Stevens, Wenbin Wu, Tanya Ivanova, Albena Stoyanova, Anatoli Dzhurmanski.

**Methodology:** Valtcho D. Zheljazkov, Ivanka B. Semerdjieva, Daniela Borisova, Elina Yankova-Tsvetkova, Lyubka H. Koleva-Valkova, Galya Petrova, Ivayla Dincheva, Fred Stevens, Wenbin Wu, Tess Astatkie, Albena Stoyanova.

**Project administration:** Valtcho D. Zheljazkov.

**Resources:** Valtcho D. Zheljazkov, Ivanka B. Semerdjieva, Daniela Borisova, Elina Yankova-Tsvetkova, Lyubka H. Koleva-Valkova, Galya Petrova, Ivayla Dincheva, Fred Stevens, Wenbin Wu, Tess Astatkie, Albena Stoyanova.

**Software:** Tess Astatkie.

**Supervision:** Valtcho D. Zheljazkov.

**Validation:** Valtcho D. Zheljazkov, Ivayla Dincheva, Fred Stevens, Tess Astatkie.

**Visualization:** Fred Stevens, Wenbin Wu, Tess Astatkie.

**Writing – original draft:** Valtcho D. Zheljazkov.

**Writing – review & editing:** Ivanka B. Semerdjieva, Daniela Borisova, Elina Yankova-Tsvetkova, Lyubka H. Koleva-Valkova, Galya Petrova, Ivayla Dincheva, Fred Stevens, Wenbin Wu, Tess Astatkie, Tanya Ivanova, Albena Stoyanova, Anatoli Dzhurmanski.

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
