## [Decision Letter · Decision Letter 0]

25 Sep 2023

PONE-D-23-25651Phytochemical and biological investigations on Centranthus kellereri (Stoj., Stef. & T. Georgiev) Stoj. & Stef. and C. ruber (L.) DC. and their potential as new medicinal and ornamental plantsPLOS ONE

Dear Dr. Zheljazkov,

Thank you for submitting your manuscript to PLOS ONE. After careful consideration, we feel that it has merit but does not fully meet PLOS ONE’s publication criteria as it currently stands. Therefore, we invite you to submit a revised version of the manuscript that addresses the points raised during the review process.

We look forward to receiving your revised manuscript.

Kind regards,

Jorddy N. Cruz

Academic Editor

PLOS ONE

Journal Requirements:

Additional Editor Comments:

Dear authors,

All corrections made to the manuscript must be marked in yellow and a point-by-point answer to all questions must be provided.

Reviewers' comments:

Reviewer's Responses to Questions

**Comments to the Author**

1. Is the manuscript technically sound, and do the data support the conclusions?

Reviewer #1: Yes

Reviewer #2: Yes

Reviewer #3: Yes

2. Has the statistical analysis been performed appropriately and rigorously? 

Reviewer #1: Yes

Reviewer #2: Yes

Reviewer #3: Yes

3. Have the authors made all data underlying the findings in their manuscript fully available?

Reviewer #1: Yes

Reviewer #2: Yes

Reviewer #3: Yes

4. Is the manuscript presented in an intelligible fashion and written in standard English?

Reviewer #1: Yes

Reviewer #2: Yes

Reviewer #3: Yes

5. Review Comments to the Author

Reviewer #1: Many correction in text.

1. Abstract clear but some corrections'

2. Introduction has to long and many corrections in text.

3. Material and methods have manty corrections in text.

4. Results have many corrections in text.

5. Discussion is clear but it has many corrections in text.

6. Conclusion is clear

7. References please check again follow the journal

Good luck

Reviewer #2: This manuscript entitled “Phytochemical and biological investigations on Centranthus kellereri (Stoj., Stef. & T. Georgiev) Stoj. & Stef. and C. ruber (L.) DC. and their potential as new medicinal and ornamental plants” described phytochemical, microstructural differences, and genetic differences of C. kellereri individuals from the two locations and biological activity of . The manuscript had the interesting subject and I think the manuscript is good for publication in this journal

Reviewer #3: The manuscript makes important contributions to the phytochemical and botanical knowledge of the species studied. It is a well-structured article, with cohesive ideas and well-presented arguments that effectively validate the results obtained. Here are some recommendations for improving the manuscript.

1) The introduction contains a lot of relevant information about the species and the importance of the study, but it is very long and makes for tiresome reading. I suggest that the author be more succinct in the introduction of the manuscript.

2) In line 110, the author says that "The closely related species Centranthus ruber DC has been extensively studied, with several reports on its bioactivity and the potential health benefits of its constituents". Reading the passage marked in red gives the impression that the plant offers benefits to its constituents and not to humans. I suggest that the author rewrite the passage to remove the ambiguity.

3) The methodology of the manuscript contains many details, including the preparation of the solutions. In my opinion, there is no need to describe the preparation of the reagents, since the methods used in the article are very well described in the literature. I recommend removing the description of the preparation of the reagents to make the methodology more summarized and more pleasant to read.

4) I noticed that (2E,6E), n-hexadecoic acid was identified in the EO extracted with hexane. I suggest revising the identification of the sample, as it is unlikely that the plant can synthesize a fatty acid with trans stereochemistry.

5) Lines 434 and 470 and tables 5 and 8 contain the term "sesqoiterpenoids". The correct spelling is "sesquiterpenoids".

6) The pictures are of poor quality, I suggest improving them and resending them.

6. PLOS authors have the option to publish the peer review history of their article (what does this mean?). If published, this will include your full peer review and any attached files.

Reviewer #1: No

Reviewer #2: **Yes: **Dr. Mohammad Moghaddam

Reviewer #3: No

---

## [Author Response · Author response to Decision Letter 0]

6 Oct 2023

Response to Reviewers

(Please note our responses are after each of the comments starting with "Response")

PONE-D-23-25651

Phytochemical and biological investigations on Centranthus kellereri (Stoj., Stef. & T. Georgiev) Stoj. & Stef. and C. ruber (L.) DC. and their potential as new medicinal and ornamental plants

PLOS ONE

Dear Dr. Zheljazkov,

Thank you for submitting your manuscript to PLOS ONE. After careful consideration, we feel that it has merit but does not fully meet PLOS ONE’s publication criteria as it currently stands. Therefore, we invite you to submit a revised version of the manuscript that addresses the points raised during the review process.

We look forward to receiving your revised manuscript.

Kind regards,

Jorddy N. Cruz

Academic Editor

PLOS ONE

Journal Requirements:

Additional Editor Comments:

Dear authors,

All corrections made to the manuscript must be marked in yellow and a point-by-point answer to all questions must be provided.

Reviewers' comments:

Reviewer's Responses to Questions

Comments to the Author

1. Is the manuscript technically sound, and do the data support the conclusions?

Reviewer #1: Yes

Reviewer #2: Yes

Reviewer #3: Yes

2. Has the statistical analysis been performed appropriately and rigorously? 

Reviewer #1: Yes

Reviewer #2: Yes

Reviewer #3: Yes

3. Have the authors made all data underlying the findings in their manuscript fully available?

Reviewer #1: Yes

Reviewer #2: Yes

Reviewer #3: Yes

4. Is the manuscript presented in an intelligible fashion and written in standard English?

Reviewer #1: Yes

Reviewer #2: Yes

Reviewer #3: Yes

5. Review Comments to the Author

Reviewer #1: Many correction in text.

Response: Because most notes relate to citing authors, we have inserted authors where necessary.

1. Abstract clear but some corrections'

Response: We corrected all plant name in Italic.

2. Introduction has to long and many corrections in text.

Response: We carefully checked the text and made corrections as necessary. Also, we have shortened the introduction to make the article easier to read. 

3. Material and methods have manty corrections in text.

Response: We have shortened the part of section M&M so that the methods are repeatable and the paper is easy to read. 

Results have many corrections in text.

Response: We corrected the Latin name of the species and cited authors as suggested. 

5. Discussion is clear but it has many corrections in text.

Response: We corrected the Latin name of the species and cited authors as suggested.

6. Conclusion is clear

Response: Thank you

7. References please check again follow the journal

Response: We checked all authors. They are formatted according to the Journals requirements.

Good luck

Reviewer #2: This manuscript entitled “Phytochemical and biological investigations on Centranthus kellereri (Stoj., Stef. & T. Georgiev) Stoj. & Stef. and C. ruber (L.) DC. and their potential as new medicinal and ornamental plants” described phytochemical, microstructural differences, and genetic differences of C. kellereri individuals from the two locations and biological activity of . The manuscript had the interesting subject and I think the manuscript is good for publication in this journal

Response:Thank you!

Reviewer #3: The manuscript makes important contributions to the phytochemical and botanical knowledge of the species studied. It is a well-structured article, with cohesive ideas and well-presented arguments that effectively validate the results obtained. Here are some recommendations for improving the manuscript.

1) The introduction contains a lot of relevant information about the species and the importance of the study, but it is very long and makes for tiresome reading. I suggest that the author be more succinct in the introduction of the manuscript.

Response: We have shortened the introduction to make the article easier to read.

2) In line 110, the author says that "The closely related species Centranthus ruber DC has been extensively studied, with several reports on its bioactivity and the potential health benefits of its constituents". Reading the passage marked in red gives the impression that the plant offers benefits to its constituents and not to humans. I suggest that the author rewrite the passage to remove the ambiguity.

Response:We rewrite this sentence as follows: “The phytochemical composition of C. ruber has been extensively studied [7-11]. There were several reports on its compositions, bioactivity and potential health benefits [7-11].

3) The methodology of the manuscript contains many details, including the preparation of the solutions. In my opinion, there is no need to describe the preparation of the reagents, since the methods used in the article are very well described in the literature. I recommend removing the description of the preparation of the reagents to make the methodology more summarized and more pleasant to read.

Response: Thank you! We shortened methodology where described the preparation of the reagents as suggested.

4) I noticed that (2E,6E), n-hexadecoic acid was identified in the EO extracted with hexane. I suggest revising the identification of the sample, as it is unlikely that the plant can synthesize a fatty acid with trans stereochemistry.

Response: Thanks you very much for your comment. In agreement with this comment, we deleted “(E,E)-“ and left only “Geranyl linalool” in the Table and throughout the text. 

Also, you noticed this: “Table 9. Chemical profile of the hexane-extracted samples: Mean concentration (%) of 3-methylbutanoic acid (isovaleric acid) (3-Meth-Iso), 3-methylpentanoic acid (3-methylvaleric acid) (3-Meth-3-M), (2E)-tridecen-1-al (2E), n-tetradecanal (n-Tetrad), (2E,6E)-farnesyl acetate (2E,6E), n-hexadecoic acid (n-Hex), and (Z,Z)-9,12-octadecadienoic acid (Z,Z)”, in response, perhaps there is a confusion with the fact that these are two different compounds: (2E,6E)-farnesyl acetate (2E,6E) and n-hexadecoic acid (n-Hex). The name of the acid is misspelled. It should be read n-hexadecanoic acid (n-Hex). It is actually palmitic acid. 5) Lines 434 and 470 and tables 5 and 8 contain the term "sesqoiterpenoids". The correct spelling is "sesquiterpenoids".

Response: Corrected

6) The pictures are of poor quality, I suggest improving them and resending them.

Response: We improved the pictures of the embryological part of this research. 

6. PLOS authors have the option to publish the peer review history of their article (what does this mean?). If published, this will include your full peer review and any attached files.

Do you want your identity to be public for this peer review? For information about this choice, including consent withdrawal, please see our Privacy Policy.

Reviewer #1: No

Reviewer #2: Yes: Dr. Mohammad Moghaddam

Reviewer #3: No

---

## [Decision Letter · Decision Letter 1]

23 Oct 2023

Phytochemical and biological investigations on Centranthus kellereri (Stoj., Stef. & T. Georgiev) Stoj. & Stef. and C. ruber (L.) DC. and their potential as new medicinal and ornamental plants

PONE-D-23-25651R1

Dear Dr. Zheljazkov,

We’re pleased to inform you that your manuscript has been judged scientifically suitable for publication and will be formally accepted for publication once it meets all outstanding technical requirements.

Kind regards,

Jorddy Neves Cruz

Academic Editor

PLOS ONE

Additional Editor Comments (optional):

The authors made the necessary modifications, so the manuscript can be accepted for publication in this journal.

Reviewers' comments:

Reviewer's Responses to Questions

**Comments to the Author**

1. If the authors have adequately addressed your comments raised in a previous round of review and you feel that this manuscript is now acceptable for publication, you may indicate that here to bypass the “Comments to the Author” section, enter your conflict of interest statement in the “Confidential to Editor” section, and submit your "Accept" recommendation.

Reviewer #3: All comments have been addressed

2. Is the manuscript technically sound, and do the data support the conclusions?

Reviewer #3: Yes

3. Has the statistical analysis been performed appropriately and rigorously? 

Reviewer #3: Yes

4. Have the authors made all data underlying the findings in their manuscript fully available?

Reviewer #3: Yes

5. Is the manuscript presented in an intelligible fashion and written in standard English?

Reviewer #3: Yes

6. Review Comments to the Author

Reviewer #3: (No Response)

7. PLOS authors have the option to publish the peer review history of their article (what does this mean?). If published, this will include your full peer review and any attached files.

Reviewer #3: No

---

## [Editor Report · Acceptance letter]

27 Oct 2023

PONE-D-23-25651R1 

Phytochemical and biological investigations on *Centranthus kellereri* (Stoj., Stef. & T. Georgiev) Stoj. & Stef. and C. ruber (L.) DC. and their potential as new medicinal and ornamental plants 

Dear Dr. Zheljazkov:

I'm pleased to inform you that your manuscript has been deemed suitable for publication in PLOS ONE. Congratulations! Your manuscript is now with our production department. 

Kind regards, 

on behalf of

Dr. Jorddy Neves Cruz 

Academic Editor

PLOS ONE